# TNFα Activates the Liver X Receptor Signaling Pathway and Promotes Cholesterol Efflux from Human Brain Pericytes Independently of *ABCA1*

**DOI:** 10.3390/ijms24065992

**Published:** 2023-03-22

**Authors:** Shiraz Dib, Rodrigo Azevedo Loiola, Emmanuel Sevin, Julien Saint-Pol, Fumitaka Shimizu, Takashi Kanda, Jens Pahnke, Fabien Gosselet

**Affiliations:** 1Blood-Brain Barrier Laboratory (LBHE), UR 2465, University of Artois, F-62300 Lens, France; shiraz_deeb@hotmail.com (S.D.); emmanuel.sevin@univ-artois.fr (E.S.); julien.saintpol@univ-artois.fr (J.S.-P.); 2Department of Neurology and Clinical Neuroscience, Graduate School of Medicine, Yamaguchi University, Ube 755-8505, Japan; 3Department of Pathology, Section of Neuropathology, Translational Neurodegeneration Research and Neuropathology Lab, University of Oslo, Oslo University Hospital, Sognsvannsveien 20, 0372 Oslo, Norway; jens.pahnke@medisin.uio.no; 4Pahnke Lab (Drug Development and Chemical Biology), Lübeck Institute of Experimental Dermatology (LIED), University of Lübeck, University Medical Center Schleswig-Holstein, Ratzeburger Allee 160, 23538 Lübeck, Germany; 5Department of Pharmacology, Faculty of Medicine, University of Latvia, Jelgavas iela 3, 1004 Riga, Latvia; 6Department of Neurobiology, The Georg S. Wise Faculty of Life Sciences, Tel Aviv University, Tel Aviv 6997801, Israel

**Keywords:** brain pericytes, TNFα, cholesterol metabolism, *ABCA1*, *LXR*, *ABCB1*

## Abstract

Neuroinflammation and brain lipid imbalances are observed in Alzheimer’s disease (AD). Tumor necrosis factor-α (TNFα) and the liver X receptor (*LXR*) signaling pathways are involved in both processes. However, limited information is currently available regarding their relationships in human brain pericytes (HBP) of the neurovascular unit. In cultivated HBP, TNFα activates the *LXR* pathway and increases the expression of one of its target genes, the transporter ATP-binding cassette family A member 1 (*ABCA1*), while *ABCG1* is not expressed. Apolipoprotein E (*APOE*) synthesis and release are diminished. The cholesterol efflux is promoted, but is not inhibited, when *ABCA1* or *LXR* are blocked. Moreover, as for TNFα, direct *LXR* activation by the agonist (T0901317) increases *ABCA1* expression and the associated cholesterol efflux. However, this process is abolished when *LXR*/*ABCA1* are both inhibited. Neither the other ABC transporters nor the SR-BI are involved in this TNFα-mediated lipid efflux regulation. We also report that inflammation increases *ABCB1* expression and function. In conclusion, our data suggest that inflammation increases HBP protection against xenobiotics and triggers an *LXR*/*ABCA1* independent cholesterol release. Understanding the molecular mechanisms regulating this efflux at the level of the neurovascular unit remains fundamental to the characterization of links between neuroinflammation, cholesterol and HBP function in neurodegenerative disorders.

## 1. Introduction

The brain is the most cholesterol-rich organ of the body and central nervous system (CNS) cholesterol synthesis accounts for the highest production rates across the body [1]. Cholesterol is an essential component of all mammalian cells, ensuring proper membrane integrity, fluidity, and biochemical function, thus affecting endocytosis, and is enriched in the myelin sheath. The membrane cholesterol pool is also important in host defense processes during immune responses [2]. In CNS, almost all of the cholesterol is produced de novo by astrocytes due to the presence of the blood–brain barrier (BBB) which strictly regulates nutrients, waste products and cell exchanges between CNS and the bloodstream [1,3]. Cholesterol is then transferred from the plasma membrane of astrocytes by the ATP-binding cassette A subfamily 1 (*ABCA1*) transporter to form high density lipoproprotein (HDL) particles comprising apolipoproteins such as *APOA-I* or *APOE*. *ABCG1*, a second ABC transporter, is then able to transfer additional cholesterol molecules from astrocyte membranes to HDL to form mature HDL that are then shuttled to neurons and taken up by the low density lipoprotein receptor (LDLR) and the LDLR-related protein 1 (LRP1) in order to be used for synaptogenesis, myelin sheath synthesis and membrane repair [4,5]. Cerebral cholesterol levels of neural cells are strictly sensed and regulated by the nuclear liver X receptors (*LXRs*) that tightly control the expression of *ABCA1* [6]. The *LXR*/*ABCA1* axis is a very promising therapeutic target for Alzheimer’s disease (AD), wherein dysfunction in CNS cholesterol metabolism and the abnormal accumulation of toxic amyloid β (Aβ) peptides have been reported. In this regard, treatments with *LXR* agonists or overexpression of *ABCA1* lead to a decrease in Aβ burden in mouse brains, which was correlated with a partial recovery of cognitive function [7,8,9]. On the contrary, deletion of *ABCA1* is associated with increased brain Aβ deposition [10,11]. APP/PS1 *LXR* null mice show increased lipid in the brain as well as Aβ deposition [12]. Furthermore, it has been reported that *ABCA1*-mediated cholesterol efflux is impaired in patients with AD and mild cognitive impairment [13]. More recently, genome-wide association studies (GWAS) have identified *ABCA1* polymorphisms as strongly linked to a high risk of developing AD [14], thus reinforcing the hypothesis that the *LXR*/*ABCA1* axis could be a promising drug target for AD. Despite these findings, the role of the *LXR*/*ABCA1* axis in the CNS remains to be clarified, in particular for CNS cholesterol homeostasis. Closely linked with the cholesterol metabolism, neuroinflammation is also a key component of AD and a recent GWAS identified the tumor necrosis factor α (TNFα) signaling pathway as a central genetic etiology of AD and related dementias [15]. However, few studies have investigated the links between TNFα and *LXR*/*ABCA1* axis [16,17].

Besides astrocytes, CNS pericytes might also play an important role in brain cholesterol metabolism and AD. However, there is a persisting gap in the knowledge of the role of brain pericytes in inflammatory conditions. These cells are embedded in the basal lamina of the brain capillaries forming the BBB and play a key role in BBB permeability and physiology [18,19,20]. During the last 15 years, advances in high resolution imaging techniques, transcriptomic methods, new cell isolation and cultivation protocols have allowed an exponentially increasing number of studies focusing on brain pericytes [21], thus highlighting new functions for them in CNS homeostasis and BBB functioning. Loss or functional defect of pericytes has been reported in AD [22]. Despite the fact that CNS inflammation is primarily driven by microglia, astrocytes and infiltrating leukocytes, important roles of brain pericytes in immune responses have also been suggested [21]. When activated by TNFα or other inflammatory stimuli, brain pericytes facilitate the trafficking of immune cells to inflammatory sites [23] and show altered phagocytic activities [24,25]. Bovine brain pericytes express *ABCA1*, but not *ABCG1*, and generate HDL when stimulated with *LXR* agonists [26]. Cholesterol efflux is a key event in immune response, as intensively studied in leukocytes and macrophages [2], but this process remains poorly studied in human brain pericytes.

Based on all of these observations, and on our previous works obtained in bovine pericytes [26,27], we therefore hypothesized that the *LXR*/*ABCA1* axis also controls the cholesterol efflux in human brain pericytes (HBP) and that this process might be affected by inflammation. Using a cell line recapitulating the HBP phenotype [28,29,30], the objective of our study was to characterize the effects of TNFα signaling on cholesterol transport/metabolism in particular focusing on the *LXR*/*ABCA1* axis.

## 2. Results

### 2.1. TNFα Promotes Expression of Inflammatory Markers and Does Not Affect Human Brain Pericytes Survival

We first assessed the effects of TNFα on HBP survival using two distinct methods. As shown in Figure 1A,B, exposure to TNFα at 5 ng/mL and 10 ng/mL for 24 h and 48 h did not affect HBP viability, indicating that this inflammatory molecule did not induce cell death at these concentrations in our culture conditions. The levels of inflammatory markers were then quantified by Western blot. Figure 1C,D show that expression of vascular cell adhesion molecule-1 (VCAM-1), cyclooxygenase 2 (COX-2) and the inflammasome molecule NOD-like receptor family, pyrin domains-containing 3 (NLRP3), were significantly upregulated after 24 h and 48 h of treatment with 5 ng/mL or 10 ng/mL of TNFα. Moreover, secretion of the pro-inflammatory interleukin 6 (IL-6) was also increased in an HBP supernatant treated with different concentrations of TNFα (Figure 1E,F). We then investigated the activation of the TNFα signaling pathway by Western blot, measuring the phosphorylation states of the nuclear factor κappa B protein (NF-κB). Figure 1G,H show a fast but brief increase in the detection of the phosphorylated NF-κB during the first 30 min after TNFα treatment. Overall, these results demonstrate that TNFα treatment triggers the NF-κB signaling pathway in HBP and induces a fast inflammatory response without altering cell viability, even during long incubation periods (24 h and 48 h).

### 2.2. TNFα Alters HBP Cholesterol Metabolism

The effect of inflammatory stimuli such as TNFα on cholesterol metabolism has been previously investigated in human macrophages and monocytes [31,32] but has never been studied in HBP. Therefore, we investigated the expression levels of several key players of the cholesterol metabolism in HBP treated with 5 and 10 ng/mL of TNFα for 24 and 48 h. We first assessed the expression levels of receptors involved in lipoproteins uptake: LDLR and LRP1. mRNA and protein levels of LRP1 were downregulated after 24 h and 48 h of TNFα treatment (Figure 2A–C). LDLR expression remained unchanged at 24 h but was significantly increased at 48 h (Figure 2A–C). 3-hydroxy-3-methylglutaryl-CoA reductase (*HMGCR*) is the rate-limiting enzyme in mammalian cells responsible for the cholesterol synthesis. No changes have been reported on mRNA and protein levels with the TNFα treatment (Figure 2A–C). In addition, we assessed the mRNA level of MYLIP, which is responsible for LDLR degradation and implicated in the control of intracellular cholesterol levels. TNFα treatments do not modulate its expression (Appendix A). Interestingly, we observed that TNFα treatment decreased the expression levels and the secretion of the cholesterol acceptor *APOE*, which is the major lipid acceptor in CNS and is the strongest genetic risk factor involved in AD [33,34] (Figure 2A–C). We then quantified the intracellular pool of cholesterol using two different methods and observed a decrease after 48 h of TNFα treatment, but not at 24 h (Figure 2D,E). These data suggest that TNFα affects the cholesterol metabolism in HBP at 48 h, without modifying the cholesterol synthesis, but rather by modulating the cholesterol efflux as demonstrated by the downregulation of the *APOE* secretion.

### 2.3. TNFα Activates the LXR Signaling Pathway

In order to gain insights into the molecular mechanisms modifying the intracellular storage and renewal of cholesterol, we studied the effects of TNFα on the liver X Receptor (*LXR*) signaling pathway that senses the intracellular cholesterol concentration and controls the transcription of the key player in cholesterol efflux, namely the ATP-binding cassette sub-family A member 1 (*ABCA1*) transporter. HBP expresses both *LXR* isoforms with *LXRα*, and is itself expressed almost three-fold more than *LXRβ* (Appendix A). TNFα receptor 1 (TNFRSF1A) is highly expressed whereas expression of TNFα receptor 2 (TNFRSF1B) is very low (Appendix A). HBP were transfected with an *LXR* reporter vector and were then treated with TNFα as previously done. No changes in the Firefly/Renilla luminescence were observed during the first 16 h of TNFα treatment (Figure 3A,B). However, a 1.5-fold increase was reported at 24 h, and a four to five-fold increase was observed at 48 h when compared with the untreated conditions. These results suggest a slight activation of the *LXR* pathway after 24 h of TNFα treatment, which was then strengthened after 48 h of treatment. No change in mRNA expression of both *LXR* isoforms was observed after TNFα treatment (Appendix A). *LXR* pathway activation was faster (at 8 h) and higher in transfected HBP treated with T0901317 (10 µM), an *LXR* synthetic agonist (Saint-Pol et al., 2012, 2013), used as a positive control in our experiments. These data strongly reinforce our hypothesis that the HBP cholesterol metabolism is altered in inflammatory situations. Interestingly, they also indicate that TNFα triggered the *LXR* signaling pathway later, after 24 and 48 h of treatment, when compared with a direct *LXR* activation by agonist.

### 2.4. TNFα Increases Expression of ABCA1 at the Transcriptional and Protein Levels

As mentioned above, *ABCA1* is the key transporter promoting cholesterol efflux from cell membranes [6]. Because its expression is tightly controlled by the *LXR* signaling pathway, we next investigated *ABCA1* expression in HBP after TNFα treatment. RT-qPCR analysis of *ABCA1* showed that TNFα treatment (5 and 10 ng/mL) upregulated mRNA expression by two-fold at 24 h, and by three–four-fold at 48 h (Figure 4A). In line with the mRNA analysis, our Western blot results also show the highest upregulation of *ABCA1* protein levels occurring after 48 h of TNFα treatment (Figure 4B,C).

To confirm that *ABCA1* upregulation is directly mediated by the *LXR* signaling pathway, we then used the *LXR* blocker, GSK2033 [35]. This inhibitor totally abolishes the *ABCA1*-upregulation mediated by TNFα and T0901317 (Figure 4D,E), thus confirming that inflammation increased *ABCA1* expression via the *LXR* pathway. The direct comparison of *ABCA1* expressions after TNFα and T0901317 treatments shown in Figure 4 also demonstrates that T0901317 is largely more efficient than TNFα at increasing *ABCA1* expression. Indeed, whereas TNFα increased *ABCA1* expression by 1.5-fold at 24 h, T0901317 increased *ABCA1* expression by 34-fold. At 48 h, the TNFα-mediated increase was 400% whereas the T0901317-induction of *ABCA1* was 1500% (Figure 4E). Therefore, *LXR* activation by TNFα in HBP is slower and less efficient than direct *LXR* activation by *LXR* agonist such as T0901317.

### 2.5. TNFα Increases Cholesterol Efflux

Because *ABCA1* is known to induce lipid efflux in many cell types in the presence of lipid acceptors such APOA-I and high-density lipoproteins (HDL) [7,26], we checked in our next experiments if TNFα finally promotes lipid efflux through *ABCA1*. Herein, we investigated the TNFα effect on cholesterol efflux in HBP (Figure 5). We first observed that HBP are more prone to release cholesterol to the lipid acceptors HDL than to APOA-I (for example, after one hour of cholesterol efflux 1.95% ± 0.16 versus 4.81% ± 0.30, respectively, Figure 5A,C) as previously reported in bovine brain pericytes [26]. The cholesterol efflux assay showed no difference between HBP treated and untreated with TNFα for 24 h in the presence of HDL and APOA-I (Figure 5A and Figure 5C, respectively). However, HBP released high amounts of cholesterol to both lipid acceptors after 48 h of treatment when compared with the untreated condition (Figure 5B,D). The efflux was even greater to HDL comparing with the efflux to APOA-I, (after 8 h of efflux, 18.17% ± 0.69 versus 5.69% ± 0.16, respectively). These findings correlate with the *ABCA1* upregulation reported earlier in Figure 4, suggesting that TNFα may increase cholesterol efflux through *ABCA1* upregulation at 48 h, but not at 24 h.

### 2.6. ABCA1 Inhibition Does Not Rescue the TNFα Induced Cholesterol Efflux

In order to establish a causal link between *ABCA1* upregulation and the increased cholesterol efflux observed after TNFα treatment in Figure 5, we next performed cholesterol efflux assays in the presence of GSK2033, an inhibitor of the *LXR* signaling pathway. Because it has been demonstrated that *LXR* activation by T0901317 also promoted cholesterol efflux to the lipid acceptors—APOE2 and APOE4—that are strongly involved in AD, we included them in the experimental set-up [26,36].

We earlier demonstrated that GSK2033 suppressed *ABCA1* expression in T0901317-treated HBP (Figure 4). In line with this result, GSK2033 suppressed the T0901317-induced cholesterol efflux (Figure 6A) confirming that the lipid release is induced through the *LXR*/*ABCA1* axis. However, GSK2033 was not able to reduce the TNFα-induced cholesterol efflux (Figure 6B), while *ABCA1* expression is decreased (Figure 4). These data suggest that TNFα and T0901317 do not share the same cholesterol efflux mechanism, and that TNFα induces an *LXR*/*ABCA1*-independent cholesterol release.

To confirm that *ABCA1* is not involved in the cholesterol release from HBP in inflammatory conditions, we then directly blocked *ABCA1* activity using Probucol, an *ABCA1* inhibitor [26,37]. Again, we observed that the inhibition of *ABCA1* decreased the cholesterol efflux after T0901317 stimulation (Figure 6C), but not when mediated by TNFα (Figure 6D). Altogether, these data reinforce the results observed in Figure 6A,B and strongly suggest that, despite the activation of the *LXR*/*ABCA1* axis observed in the presence of TNFα, the cholesterol efflux to acceptors (*HDL*, *APOA-I*, *APOE2*, *APOE4*) is controlled by another or other transporter(s).

### 2.7. TNFα Modifies the Expression of Other Transporters Involved in Lipid Efflux

We demonstrated that TNFα induces an *ABCA1*-independent cholesterol efflux in HBP. To identify which transporter is responsible for this efflux, we investigated the gene and protein expression of other transporters that might be involved in lipid efflux, such as the ABC transporters *ABCG1*, *ABCG4*, *ABCB1* (P-gp) and scavenger receptor class B member 1 (SR-BI). As with *ABCA1*, *ABCG1* is known to be regulated by the *LXR* pathway and promotes cholesterol release to HDL [38]. However, its expression has not previously been observed in bovine brain pericytes and human fibroblasts [27,39,40]. SR-BI has previously been shown to be expressed by brain pericytes [27] but its function has never been studied. The SR-BI protein is encoded by the *SCARB1* gene that is not controlled by the *LXR* signaling pathway. SR-BI is considered to be a receptor for HDL and thus promotes bidirectional cholesterol exchanges between lipoproteins and the membranes of cells [41]. *ABCG4* is highly expressed in the brain, shows high amino acid similarity with *ABCG1* and is involved in cholesterol efflux to HDL in neurons and other cell types [42]. *ABCB1* is a well-known efflux pump expressed by several cell types of the body, usually present at the physiological barriers (testis, BBB, intestinal barrier, etc.), and able to release xenobiotics from the plasma membrane into the bloodstream. *ABCB1* was initially described as a lipid flippase, able to transport hydrophobic lipids from the inner to the outer leaflet of the membrane [43,44]. It has also been suggested that lipid composition of the membrane could affect its functionality, thus proposing an interesting mechanism to modulate its protecting activity. 

Despite our effort by testing several primers pairs and antibodies, our RT-qPCR and Western blot tests suggest that *ABCG1* is not expressed in HBP and is thus in line with previous data obtained in bovine pericytes [26]. T0901317 and TNFα did not induce its expression (Appendix A and Figure 7A and Figure 7C, respectively). No significant variation of *ABCG4* expression was observed after 24 and 48 h of TNFα treatment (Figure 7A,B). Interestingly, mRNA and protein expressions of *ABCB1* were upregulated after 48 h of treatment. While our RT-qPCR analysis for *SCARB1* did not show any variation at the gene expression level (Figure 7A), our Western blot analysis identified an additional signal for SR-BI that appeared only 48 h after TNFα treatment (Figure 7B,C). However, the molecular weight of this band is lower than the expected molecular weight reported for SR-BI. To discard any doubts about *SCARB1* implication, we considered *SCARB1*, as well as *ABCB1*, to be our potential suspects responsible for the observed TNFα-induced cholesterol efflux.

### 2.8. ABCB1 and SR-BI Inhibition Does Not Rescue the TNFα-Induced Cholesterol Efflux

To determine if SR-BI and *ABCB1* might be responsible for the increased cholesterol efflux observed in HBP treated with TNFα, we next performed radiolabeled cholesterol efflux assays in the presence of the *ABCB1* inhibitors verapamil and elacridar [45,46], and an SR-BI inhibitor, BLT-1 [47]. Our results show that inhibitors did not decrease the cholesterol release, thus discarding the idea of the contribution of SR-BI and *ABCB1* in this process (Figure 8). These results definitively rule out the possible involvement of SR-BI and *ABCB1* in cholesterol release to lipid acceptors.

### 2.9. Drug Efflux Activity of ABCB1 Is Promoted by TNFα

At the BBB, *ABCB1* is expressed by endothelial cells and brain pericytes and protects the CNS from harmful substances [27,40]. It has been previously demonstrated that TNFα modifies *ABCB1* expression and functionality in BBB endothelial cells [48]. For the first time, we report that TNFα increased *ABCB1* expression in HBP (Figure 7). To determine if the TNFα promotes *ABCB1* functionality, we next assessed the efflux drug activity of *ABCB1* using the pump out system. This method consists of accumulating within cells an *ABCB1* substrate (rhodamine 123) and subsequently monitoring the *ABCB1* mediated release of this compound in real time [46]. The same inhibitors that we used above (Figure 8) are included, as well as diazepam as a negative control because this molecule is a non-*ABCB1* transported drug [46]. The kinetic of rhodamine 123 release shown in Figure 9A confirms that TNFα treatment significantly increased the efflux pump activity of the *ABCB1* at 48 h but not at 24 h. The calculated K_out_ of rhodamine 123 at 48 h is increased (130.68% ± 2.73% versus 100.00% ± 1.87%) in control conditions. This TNFα-increased K_out_ is not modified in the presence of diazepam (129.71% ± 3.29%) and is significantly decreased in the presence of verapamil and elacridar, which inhibit *ABCB1* activity (Figure 9B, 80.41% ± 2.66% and, 98.94% ± 2.71%, respectively). These results clearly demonstrate that elacridar and verapamil, also used in Figure 8, efficiently inhibit P-gp functions. This strengthens our previous conclusion that P-gp is not involved in cholesterol efflux after TNFα stimulation.

Because the transporter breast cancer cells resistant protein (BCRP, aka ABCG2) shows substrate overlap with *ABCB1*, as we previously observed in other cell types [45,46], we also studied ABCG2 expression in HBP. Our data demonstrate that ABCG2 expression is not affected by TNFα (Appendix A). These results strongly support the notion that inflammatory stimuli such as TNFα, promote *ABCB1* expression and function in HBP.

## 3. Discussion

In mammals, cholesterol is the major component of the cell membrane where it plays numerous roles in establishing crucial biochemical and biophysical properties, including cell signaling, organization of lipid microdomains, receptor distribution and function, bilayer fluidity, and membrane integrity [49]. A new emerging concept suggests that this membrane cholesterol pool might be divided into three distinct pools, each of them involved in inflammatory responses and defenses (reviewed in [2]).

In the brain, which is the most cholesterol-rich organ of the body, cholesterol also plays a central role in myelin sheath formation, synaptogenesis and membrane repair [1]. *LXR*/*ABCA1* axis controls cholesterol homeostasis within the CNS, and disturbances in this signaling pathway might contribute to several neurological disorders such as in AD. Therefore, the *LXR*/*ABCA1* axis and ABCA transporters have been previously suggested as promising therapeutical targets in AD [50,51,52,53]. Recent identification of *ABCA1* as an important genetic factor for AD by genome wide association studies (GWAS) strengthen this hypothesis [14].

In our study, we investigated the *LXR*/*ABCA1* axis in human brain pericytes (HBP) under inflammatory situations that are also a pathological feature of several neuroinflammatory diseases, including AD [15]. Brain pericytes are mural cells, embedded in the vascular basal lamina of the brain capillaries, and participating in the blood–brain barrier (BBB) formation and maintenance [19,20]. Loss or defect of brain pericytes has been reported in AD, suggesting that they can significantly contribute to AD onset and progression [22]. Recent advances in high resolution imaging techniques, transcriptomic techniques, new cell isolation methods, and cultivation protocols allow the highlighting of new functions for brain pericytes in CNS homeostasis and BBB functioning. For example, Erdener et al. have demonstrated that brain pericytes express several actin isoforms and myosin heavy chain type 11 suggesting that such cells might regulate the cerebral blood flow due to their ability to contract in response to stimuli [54]. Brain pericytes also play a central role in CNS immune response as it has been demonstrated that TNFα promotes expression of cell surface adhesion molecules such as VCAM1, triggering trafficking of leukocytes to inflammatory sites [55]. Our results, obtained in HBP confirm that TNFα increases VCAM1 expression in vitro. We also observed an increase in IL-6 production and secretion. Other in vitro studies performed with porcine and human brain pericytes previously showed the phagocytic activity of cell debris in inflammatory conditions [24,25,56], thus strongly suggesting that brain pericytes play an essential role in neuroinflammation.

The majority of the biological processes regulating CNS acute inflammation has focused on the contribution of glial cells, which are undoubtedly crucial for host defense and cell survival. However, there is a persisting gap in the knowledge of the role of brain pericytes in inflammatory conditions. Therefore, we studied the cholesterol metabolism and the *LXR*/*ABCA1* axis using a human cell line that is very well characterized and summarized the major molecular signature of brain pericytes [28,29,30]. HBP highly express *LXRα* and, to a lower expression level (three-fold less), *LXRβ*. For the first time, we demonstrated in human brain pericytes that TNFα triggers the *LXR* signaling pathway, thus enhancing the *ABCA1* expression that correlates with a lower cholesterol intracellular accumulation and an increase of the cholesterol efflux to *HDL*, *APOA-I*, *APOE2* and *APOE4*. Activation of the *LXR* signaling pathway by synthetic agonist T0901317 gives the same trend as previously demonstrated in bovine brain pericytes [26]. However, neither *ABCA1* blockade nor *LXR* signaling pathway inhibition affected this TNFα-mediated efflux, suggesting that this process is possibly passive and independent of *LXR* and *ABCA1*. Previous studies have already observed an upregulation of *ABCA1* and of the cholesterol efflux in peritoneal mouse macrophages treated with TNFα and have attributed this lipid release to *ABCA1* [17]. However, no *LXR* or *ABCA1* inhibitions were performed. Additionally we cannot exclude the idea that the effect of TNFα is ultimately cell type dependent because human vein endothelial cells treated with TNF-α showed neither *ABCA1* upregulation nor change in the cholesterol efflux [31].

We excluded participation of other ABC transporters and scavenger receptors described as participating in cholesterol efflux such as *ABCG1*, *ABCG4* and *SCARB1*. As in bovine brain pericytes and human fibroblasts, *ABCG1* is absent in HBP [27,39] and is not upregulated by *LXR* activation or TNFα. Interestingly, *APOE* expression and secretion are also decreased after TNFα treatment whereas expressions of the LDLR and LRP1 responsible for the lipoprotein’s uptake are increased and decreased, respectively. While TNFα altered the cholesterol uptake and efflux, cholesterol synthesis is not affected. Altogether, these data suggest that TNFα affects cholesterol homeostasis in HBP by (i) modifying the uptake of lipoproteins, and (ii) promoting cholesterol efflux but independently of *ABCA1*. All these modifications lead to a decrease of the total cholesterol pool of HBP. One major observation of this study is that *ABCA1* is not involved in cholesterol efflux in inflammatory conditions. Therefore, the role of *ABCA1* in this context needs further investigations. An alternative function for *ABCA1* in macrophages may be to promote efferocytosis, which refers to the engulfment/clearance of apoptotic cells, as has been suggested by previous studies [17,57]. For instance, *ABCA1*^−/−^ mice and macrophages show an impairment of recognition and clearance of apoptotic cells [58,59]. As mentioned previously, brain pericytes show phagocytic activity [21] thus suggesting that further investigations are needed to understand if the upregulation of *ABCA1* in TNFα-treated HBP could promote cell debris clearance.

We were not able to identify another lipid transporter responsible for the increased cholesterol release observed after TNFα stimulation. This suggests that the mechanism of this cholesterol release might be mediated by another transporter as recently suggested in red blood cells by Ohkawa et al. [60]. Another explanation might be that this lipid release corresponds to the aqueous diffusion pathway in which the cholesterol molecules from the plasma membrane (PM), in contact with the HDL particles, can be trapped without the involvement of a transporter. This hypothesis is supported by an emerging concept in which the cholesterol pool of the membranes can be divided into three distinct pools [2]:The first pool is the accessible or metabolically active pool, which is usually low (around 10% of the total PM cholesterol pool). This pool can be transferred to HDL and apolipoproteins by a process involving a transporter (such *ABCA1*) or by the aforementioned process called aqueous passive diffusion. If not transferred at external sources, cholesterol molecules of this pool can also be transferred to the intracellular pool of cholesterol located in the endoplasmic reticulum (ER), which senses the total cholesterol pool of the cells. Therefore, this metabolic pool is currently suspected to be a key player in the immune response and defense processes. This pool can be targeted by bacteria and viruses in order to penetrate within cells. Interestingly, it has been suggested that brain pericytes are more permissive for viral and bacterial infection than other CNS cells [61].The second pool is the phospholipid-sequestered pool (almost 45% of the total PM pool) representing a strong biophysical basis for the assembly of lipid microdomains or lipid rafts.The last cholesterol pool is the essential pool (45% of PM pool also), currently suspected to play a fundamental role in lipid bilayer integrity.


Based on this, we hypothesize that, in HBP, TNFα increases this metabolically active pool of cholesterol of the PM that can therefore be released to HDL and apolipoproteins in the highest quantity by a process that is not mediated by *ABCA1*. It seems to be rather mediated by a passive aqueous phenomenon during which HDL interact with the PM to trap cholesterol molecules of this active pool. Consequently, there is an increase in the HDL lipidation. At the periphery, HDL have anti-inflammatory and anti-oxidative properties [62,63]. We cannot exclude that HDL particles also have similar functions in CNS to mitigate damages in neurological diseases. Further investigations are needed in this direction to understand if the HDL and HDL-like particles generated by the brain pericytes cholesterol efflux might have CNS anti-inflammatory actions.

Another important result of our study is the understanding that inflammation can upregulate *ABCB1* expression and function in HBP. We first suspected *ABCB1* to be responsible for the increased cholesterol efflux observed in the presence of TNFα but we only observed an increased involvement of this transporter in cellular drug release. To our knowledge, this is the first time that similar effects have been reported in HBP. In the endothelial cells of the BBB, we and others have reported that TNFα decreases *ABCB1* and ABCG2 expression and functionality, thus suggesting different mechanisms of regulation in these cell types [48,64]. We have previously reported the upregulation of *ABCB1* when the *LXR* signaling pathway is activated by natural or synthetic agonist in endothelial BBB cells [36]. Altogether, these data strongly suggest that TNFα activates the *LXR* signaling pathway that in turn increases *ABCB1* expression and functions.

Importantly, we generated these data in HBP cultured in vitro. Further studies with human samples or tissues are needed to confirm these effects of the inflammation at the human BBB and CNS.

## 4. Materials and Methods

### 4.1. Human Brain Pericytes (HBP)

Human brain pericytes (HBP) were isolated from a patient who had suddenly died from a heart attack [28]. Study protocol for human tissue was approved by the ethics committee of the Medical Faculty (IRB#: H18-033-6), University of Yamaguchi Graduate School, and was conducted in accordance with the Declaration of Helsinki, as amended in Somerset West in 1996. Written informed consent was obtained from the family of the participant before entering the study. The protocol was approved by the French Ministry of Higher Education and Research (CODECOH Number DC2011-1321). All experiments were carried out in accordance with the approved protocol.

Briefly, these pericytes were transfected and immortalized using retro-virus vectors holding human temperature-sensitive SV40 T antigen (tsA58) and human telomerase (Htert). HBP were then amplified and cultured at 33 °C in high glucose (4.5 g/L) Dulbecco’s modified Eagles’ medium (DMEM/HG), supplemented with 10% non-heat-inactivated fetal calf serum (FCS, Sigma-Aldrich, Saint-Louis, MS, USA), 1% L-glutamine (Merck chemicals, Darmstadt, Germany) and 1% penicillin–streptomycin (Sciencell, Carlsbad, CA, USA). Morphology and expression of HBP markers such as desmin, neuroglial2 (NG2), α smooth muscle actin (αSMA) and platelet-derived growth factor receptor (PDGF-R) were evaluated, while HBP have been deeply characterized in previous studies [28,29,30].

Before seeding, well plates were coated with collagen I (Corning, NY, USA). Then, HBP were seeded in 6-well plates at a density of 150,000 cells/well, in 12-well plates at a density of 50,000 cells/well, in 24-well plates at a density of 25,000 cells/well or in 96-well plates at 5000 cells/well for pump out assays and 7500 cells/well for *LXR* signal reporter assay. Only brain pericytes at passage 11 were used in this study. Cells were cultured at 37 °C in DMEM/HG supplemented with 1% L-glutamine (Merck KGaA, Darmstadt, Germany) and 1% penicillin–streptomycin (Sciencell) during 48 h to reach 80 to 90% of confluence.

### 4.2. Treatment of HBP

TNFα (Sigma-Aldrich) was dissolved in DMEM/HG supplemented with 0.1% bovine albumin serum (BSA, Sigma-Aldrich) at a concentration of 10 µg/mL. When ready, HBP were rinsed once with warm DMEM/HG/0.1% BSA and then incubated in DMEM/HG/0.1% BSA supplemented or not with 5 or 10 ng/mL of TNFα. These concentrations were selected based on previous in vitro studies published by us and others [64,65,66,67].

### 4.3. Cell Viability and Cytotoxicity Assessment

Cell viability was assessed for all tested molecules (TNFα, inhibitors, agonists, etc.) by performing the resazurin test and MTT assay. Please see the Appendix A for toxicity assessment of all the compounds used in this study on HBP. The resazurin assay was performed based on the protocol of Jennings et al. [68]. A resazurin stock solution at 880 µM (R7017, Sigma) was prepared by dissolving the powder in 0.1 N NaOH (0.011 g/mL) and diluting it in phosphate saline buffer (PBS) with a final pH of 7.8. At the end of 24 and 48 h of treatment with TNFα, HBP were rinsed once with warm PBS then incubated for 2 h in 500 µL of resazurin diluted 20 times in DMEM 0.1% BSA solution. Fluorescence of resorufin (resazurin reduction product) was measured using the Synergy H1 plate reader (Agilent, Santa Clara, CA, USA) (λ excitation = 540 nm & λ emission = 590 nm). The MTT tetrazolium viability assay (Sigma-Aldrich) was performed according to manufacturer instructions and using the MTT (3-(4,5-dimethylthiazol-2-yl)-2,5-diphenyltetrazolium bromide). After treatments with different compounds, or after pump out experiments, HBP were incubated for 1 h with MTT tetrazolium diluted in Ringer–HEPES (RH buffer) buffer (150 mM NaCl, 5.2 mM KCl, 2.2 mM CaCl_2_, 0.2 mM MgCl_2_, 6 H_2_O, 6 mM NaHCO_3_, 5 mM HEPES, pH: 7.4) at 37 °C. HBP were then lysed using DMSO. Formazan absorbance was then measured in treated and in control HBP using the Synergy H1 reading spectrophotometer at 570 nm.

### 4.4. RNA Isolation, Reverse Transcription and Quantitative PCR

Nucleospin RNA/Protein Kit from Macherey-Nagel (Macherey-Nagel, Dueren, Germany) was used to isolate mRNA from HBP cultures following the manufacturer protocol. Briefly, after treatments, cells were rinsed once with cold RH buffer and then lysed with RP1 10% DTT lysis buffer. RNA purity was monitored by measuring the absorbance at 260, 280 and 320 nm using a Take 3 plate and Agilent’s Synergy H1 spectrophotometer. Reverse transcription was performed using IScript Reverse Transcription Supermix (Bio-Rad, Hercules, CA, USA) and following the manufacturer’s instructions. An amount of 250 ng of mRNA was used to obtain cDNA for each sample. qPCRs were performed in 96-well plates, where to each well we added: 2 µL of obtained cDNA, 0.4 µL of forward and 0.4 µL of reverse primers (10 pmol), 2.2 µL of RNAse-free water, and 5 µL of SoFast EvaGreen Supermix (Bio-Rad). Amplification was monitored using the CFX96 thermocycler (Bio-Rad) by performing 40 cycles with an annealing temperature of 60 °C. Data were analyzed using the Bio-Rad CFX Manager software and target genes expression levels were normalized to the housekeeping gene *GAPDH* using the ΔΔCt method. Specificity and efficiency for all target genes primers were evaluated before performing qPCR. Primer sequences are listed in Table 1.

### 4.5. Protein Extraction and Immunoblots

At the end of the treatment, cells were rinsed twice with cold RH buffer and lysed with RIPA buffer (Sigma-Aldrich) supplemented with anti-proteases and -phosphatases (Sigma-Aldrich). The lysates were then centrifuged at 10,000 rpm for 10 min at 4 °C. Pellets were discarded and the protein concentration present in the supernatant was quantified using the Bradford method (Bio-Rad). An amount of 20 µg protein per sample was added to 4x Laemmli buffer (Bio-Rad) and used for immunoblotting. Depending on each antibody’s working conditions, proteins were heated or not and supplemented or not with β-mercaptoethanol (Sigma-Aldrich) (Table 2). Prepared cell lysates were loaded into acrylamide fixed concentration gels for electrophoresis (SDS-PAGE) and transferred onto a nitrocellulose membrane (Amersham Biosciences, Buckinghamshire, United Kingdom). After transfer, blocking of the membranes was performed with 5% non-fat dry milk diluted in Tris base saline-1% tween 20 (TBST) and supplemented with 3% serum normal goat (SNG, Sigma-Aldrich) for at least 1 h. Depending on the working conditions of each primary antibody, incubation was performed overnight at 4 °C or 2 h at room temperature (RT). Antibodies were diluted in TBST (Table 2). After primary antibody incubation, membranes were rinsed three times with TBST and incubated with mouse, rabbit or rat secondary antibodies coupled to horseradish peroxidase (Dako, Lostrup, Denmark) for 1 h at RT. Proteins were revealed using enhanced chemiluminescence kit (GE Healthcare, Little Chalfont, UK). Chemiluminescence was observed by Western blot imaging system Azure c600 (Azure Biosystems, Dublin, Ireland) and quantified using AzureSpot 2.0 software. Each target protein was normalized to β-actin (ACTIN). Primary antibody details are shown in Table 2. Representative images of uncropped, full blots for all targets assessed in this study are shown in Appendix A.

### 4.6. APOE and IL-6 Determination in Cell Culture Supernatant

Supernatants were collected after each TNFα treatment and stored at −20 °C. To precipitate secreted proteins, four volumes of pure acetone (Sigma-Aldrich) were added to one volume of supernatant and incubated at −20 °C overnight. After a centrifugation step at 10,000× *g* for 10 min, supernatant was discarded, and the remaining acetone was air-dried. The pellet was resuspended in RIPA lysis buffer supplemented with anti-proteases and anti-phosphatases. Bradford assay was performed for protein quantification and 50 µg of proteins were used for Western blot analysis. Then, immunoblots were performed as described above. Secreted targets were normalized to BSA.

### 4.7. Pump out Assay

The initial “pump out” method has been developed using Caco-2 cells [46]. In the present study, this method has been updated for HBP. After seeding and TNFα treatment, cells were rinsed once with warm RH buffer and incubated with 10 µM of rhodamine 123 (Sigma-Aldrich) solution during 2 h for the assessment of *ABCB1* (P-gp) functionality. After incubation period, cells were rinsed once with warm RH buffer and then incubated in RH buffer with or without the following compounds, all purchased at Sigma-Aldrich: 50 µM diazepam (control molecule because not substrate of these efflux pumps), 10 µM elacridar (*ABCB1* and ABCG2 inhibitor), 50 µM verapamil (*ABCB1* inhibitor). Fluorescent dye efflux was measured every 2 min during 60 min at 37 °C, and the efflux rate (K_out_) was calculated as the slope of the dye cumulative amount curve over time. K_out_ of treated and untreated conditions were compared for different tested inhibitors. The measurements were taken by the microplate fluorescence reader SYNERGY H1 for rhodamine 123 (λ_ex_ = 501 nm and λ_em_ = 538 nm). After each pump out experiment, MTT assay was performed to assess HBP viability after incubation with the different chemical compounds.

### 4.8. Cellular Cholesterol Efflux Assay 

Radiolabeled medium was obtained following two steps. In the first step, 0.5 µCi/mL of [^3^H]-Cholesterol (Perkin Elmer, Waltham, MA, USA) was added to FCS and incubated at 37 °C during 3 h. In the second step, radiolabeled serum was completed with DMEM/HG, 1% L-glutamine and 1% penicillin–streptomycin (completing the HBP medium). HBP were seeded as described previously in 24-well plate format. After cell attachment, medium was removed and replaced by 500 µL of the radiolabeled media for 36 h. Then, HBP were rinsed twice with DMEM/HG and incubated with DMEM/HG/0.1% BSA in the absence or presence of 10 ng/mL of TNFα for 24 and 48 h. After treatment, HBP were rinsed once with DMEM/HG/0,1% BSA and incubated for 1 h, 2 h, 4 h and 8 h with DMEM/HG/0.1% BSA containing or not the lipid acceptors: 50 µg/mL of high-density lipoproteins (HDL, Sigma-Aldrich), 20 µg/mL of apolipoprotein A-I (APOA-I, Sigma-Aldrich) or 20 µg/mL of apolipoprotein E2 or E4 (APOE2 & E4, Preprotech, Neuilly-sur-Seine, France). At the end of each time point, supernatants were collected and HBP were rinsed four times with cold RH buffer before being lysed in 1% Triton X-100 (Sigma-Aldrich). Supernatants and cell lysates were then centrifuged for 4 min at 4000 rpm. Radioactivity in the supernatant and lysates was quantified using an HIDEX 300SL scintillation counter (Sciencetec, Villebon-sur-Yvette, France) and the relative cholesterol efflux was calculated as previously described by us [26,36,69] and based on the following formula: (1)$$\mathrm{Relative}\;\mathrm{Cholesterol}\;\mathrm{Efflux}\;(\%)=100\frac{\mathrm{Supernatant}\;\mathrm{radioactivity}\;\left[\mathrm{Bq}\right]}{\mathrm{Supernatant}\;\mathrm{radioactivity}\;\left[\mathrm{Bq}\right]+\;\mathrm{Lysate}\;\mathrm{radioactivity}\;\left[\mathrm{Bq}\right]}$$

After the preliminary time points tests for cholesterol efflux estimation, we used the time point 8 h for the rest of our experiments. For inhibition experiments, we used GSK2033 (inhibitor of the *LXR* signaling pathway, MedChemExpress, Clinisciences, Nanterre, France) at 1 µM and incubated the cells 30 min before TNFα treatments. Inhibitors of ABC transporters activity were used during the 8 h of cholesterol efflux with the following concentrations: 100 µM of probucol (*ABCA1* inhibitor [37]); 10 µM elacridar (*ABCB1* inhibitor [45]); 50 µM verapamil (*ABCB1* [46]); 10 µM BLT1 [70,71]. Amounts of 10 µM of T0901317 and 1 µM of GSK2033 were used as *LXR* agonist and inhibitor, respectively, as previously described [26,35].

### 4.9. Intracellular Cholesterol Quantification 

Intracellular cholesterol dosage in HBP was performed using a cholesterol quantitation kit from Sigma-Aldrich. Briefly, HBP were seeded in 6-well plates, then treated with 10 ng/mL TNFα for 24 and 48 h, as described above, and rinsed twice with cold RH buffer. Cells were scrapped in 500 µL of RH buffer and lysates were collected. Lysates were centrifuged at 2000 rpm during 5 min and pellets were dissolved in 200 µL of chloroform:isopropanol:IGEPAL (7:11:0.1) (Sigma-Aldrich) lysis buffer. After 10 min of 10,000× *g* centrifugation, organic solvent phases were transferred in new collection tubes and pellets were discarded. Lipid extraction and cholesterol dosage were performed according to the manufacturer’s instructions. These data were then correlated with cholesterol accumulation also calculated in the cholesterol efflux experiments. 

### 4.10. Assessment of LXR Pathway Activation

HBP were plated in a 96-well plate as described above. The following day, cells were transiently transfected with the *LXR* signal reporter assay (CCS-0041L, Qiagen, Hilden, Germany) using the Jetprime transfection reagent (Polyplus, Illkirch, France). After 24 h of transfection, HBP were rinsed and treated with 5 and 10 ng/mL of TNFα or with 10 µM of T0901317 for 1, 2, 4, 8, 24 and 48 h. After each time point, HBP were rinsed once with calcium- and magnesium-free phosphate buffered saline (PBS-CMF; 8 g/L NaCl, 0.2 g/L KCl, 0.2 g/L KH_2_PO_4_, 2.86 g/L Na_2_HPO_4_·12 H_2_O; pH 7.4), and then lysed with passive lysis buffer (PLB 1X). Firefly and Renilla Luciferase luminescence was assessed using the Dual-Luciferase^®^ Reporter Assay System (E1910, Promega, France) following the manufacturer’s instructions, and using the Agilent Synergy H1 microplate reader.

### 4.11. Statistical Analysis

Results are presented as the mean ± SEM or ± SD as indicated in each figure legends. Normal distribution of values was analyzed using the Shapiro–Wilk test. Statistical testing for significance was performed using Student’s *t*-test or one-way ANOVA test followed by different tests for multiple comparison as indicated in figure legends. All the statistical tests were performed using Prism Software (GraphPad Software Inc., San Diego, CA, USA).

## 5. Conclusions

Our results are summarized in Figure 10. In this study, we observed for the first time in HBP that TNFα increases cholesterol efflux by a process independent of the *LXR*/*ABCA1* axis. No other investigated lipid transporters are involved in this process, suggesting that this lipid release is likely a passive mechanism that might play a key role in HBP resistance to stressful situations or infection. Further studies are needed to better understand this process, and to decipher the role of the *LXR*/*ABCA1* axis in HBP during neurodegenerative diseases onset and progression.

## Figures and Tables

**Figure 1 ijms-24-05992-f001:**
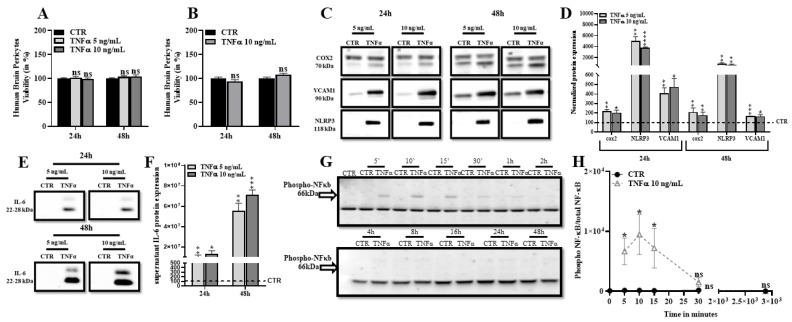
TNFα induces inflammatory markers expression but does not affect HBP viability. (**A**). After TNFα treatment, HBP were incubated with resazurin. Viable HBP converted resazurin into fluorescent Resorufin, which was measured and represented in percentage relative to the control condition (CTR). Each bar represents the mean ± SEM of three independent experiments with four replicates each (N = 3; n = 12). Statistical analysis: Student’s *t*-test, ns: non-significant. (**B**). MTT assay was performed to the highest applied dose. Data are shown as the mean ± SEM of three experiments with eight replicates each (N = 3, n = 24). Statistical analysis: Student’s *t*-test, ns: non-significant. (**C**). Cell inflammatory markers’ (*COX2*, *NLRP3*, *VCAM1*) protein levels were monitored by Western blot. (**D**). Each bar represents the level of quantified protein signal normalized to ACTIN relative to the CTR. Data are shown as the mean ± SEM of three independent experiments (N = 3). (**E**). Secreted inflammatory marker IL-6 protein level was analyzed by Western blot after supernatant proteins precipitation. (**F**). Each bar represents the level of quantified protein signal normalized to albumin relative to the CTR. Statistical analysis: Student’s *t*-test, ** p* < 0.05, ** *p* < 0.01, *** *p* < 0.001, **** *p* < 0.0001. (**G**). TNFα treatment was performed in kinetic and NF-κB pathway activation was analyzed by Western blot. (**H**). Data are shown as the mean ± SEM of three independent experiments (N = 3). Statistical analysis: Student’s *t*-test, ns: non-significant, ** p* < 0.05.

**Figure 2 ijms-24-05992-f002:**
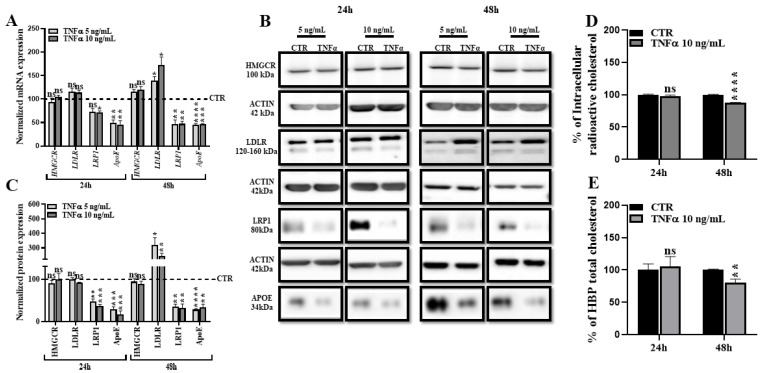
TNFα decreases HBP intracellular cholesterol level without modulating *HMGCR* expression. (**A**). Transcriptional expression of *HMGCR*, *LDLR*, *LRP1* and *APOE* was measured by RT-qPCR. Each bar represents the level of mRNA expression normalized to the housekeeping gene *GAPDH*, relative to the control condition (CTR). Data are shown as the mean ± SEM of three experiments from pooled triplicates (N = 3). Statistical analysis: Student’s *t*-test, ns: non-significant, ** p* < 0.05, ** *p* < 0.01, *** *p* < 0.001, **** *p* < 0.0001. (**B**). *HMGCR*, LDLR, LRP1 and *APOE* protein level was assessed by Western blot. Among all the investigated proteins, only *APOE* was quantified from the supernatant and normalized to albumin. (**C**). Each bar represents the level of quantified protein normalized to ACTIN, relative to the CTR. Data are shown as the mean ± SEM of three independent experiments (N = 3). Statistical analysis: Student’s *t*-test, ns: non-significant, ** p* < 0.05, ** *p* < 0.01, *** *p* < 0.001, **** *p* < 0.0001. (**D**). Total intracellular cholesterol accumulations were first monitored by using radiolabeled cholesterol as mentioned in the material and methods section. Data are shown as the mean ± SEM of 25 experiments of cholesterol efflux, representing 96 biological replicates. Statistical analysis: Student’s *t*-test, ns: non-significant, **** *p* < 0.0001. (**E**). Quantification of total intracellular cholesterol using a cholesterol quantitation kit shows reduced cholesterol at 48 h of TNFα treatment. Each bar represents the percentage of total cholesterol relative to the CTR. Data are shown as the mean ± SEM of three independent experiments within three replicates each (N = 3; n = 9). Statistical analysis: Student’s *t*-test, ns: non-significant, ** *p* < 0.01.

**Figure 3 ijms-24-05992-f003:**
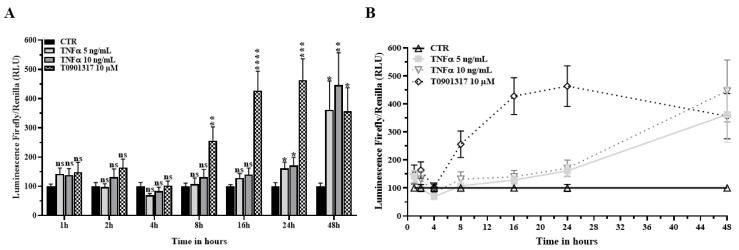
TNFα activates the *LXR* signaling pathway. (**A**). HBP were transfected with an *LXR* signal reporter assay. T0901317 was used as a positive control for *LXR* activation. Luminescence was measured after 1, 2, 4, 8, 16, 24, and 48 h of TNFα treatment. Each bar represents the mean ± SEM of three independent experiments within three replicates each (N = 3; n = 9). (**B**). Results are plotted against the control condition (CTR). Statistical analysis: Student’s *t*-test, ns: non-significant, * *p* < 0.05, ** *p* < 0.01, *** *p* < 0.001, **** *p* < 0.0001.

**Figure 4 ijms-24-05992-f004:**
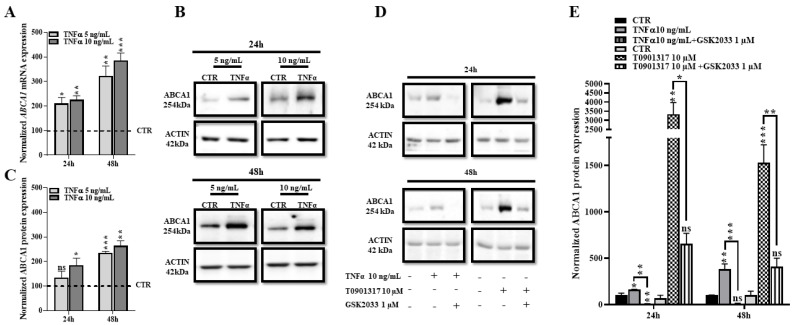
TNFα treatment increases *ABCA1* mRNA and protein expression and *LXR* inhibition alleviates TNFα-mediated *ABCA1* expression. (**A**). Transcriptional expression of *ABCA1* was measured by RT-qPCR. Each bar represents the level of mRNA expression normalized to the housekeeping gene *GAPDH*, relative to the control condition (CTR). Data are shown as the mean ± SEM of three experiments from pooled triplicates (N = 3). Statistical analysis: Student’s *t*-test, ns: non-significant, * *p* < 0.05, ** *p* < 0.01, *** *p* < 0.001. (**B**). *ABCA1* protein level was assessed by Western blot. (**C**). Each bar represents the level of quantified *ABCA1* signal normalized to ACTIN, relative to the CTR condition. Data are shown as the mean ± SEM of three independent experiments (N = 3). Statistical analysis: Student’s *t*-test, ns: non-significant ** *p* < 0.01, *** *p* < 0.001. (**D**). To inhibit *LXR* activation, HBP were pre-treated for 30 min with GSK2033 (1 µM) then incubated with T0901317 (10 µM) or with TNFα (10 ng/mL) for 24 and 48 h. (**E**). Each bar represents *ABCA1* protein expression normalized to ACTIN, relative to the control condition (CTR). Data are shown as the mean ± SEM of three independent experiments (N = 3). Statistical analysis: one-way ANOVA followed by Tukey’s multiple comparison, ns: non-significant, * *p* < 0.05, ** *p* < 0.01, *** *p* < 0.001.

**Figure 5 ijms-24-05992-f005:**
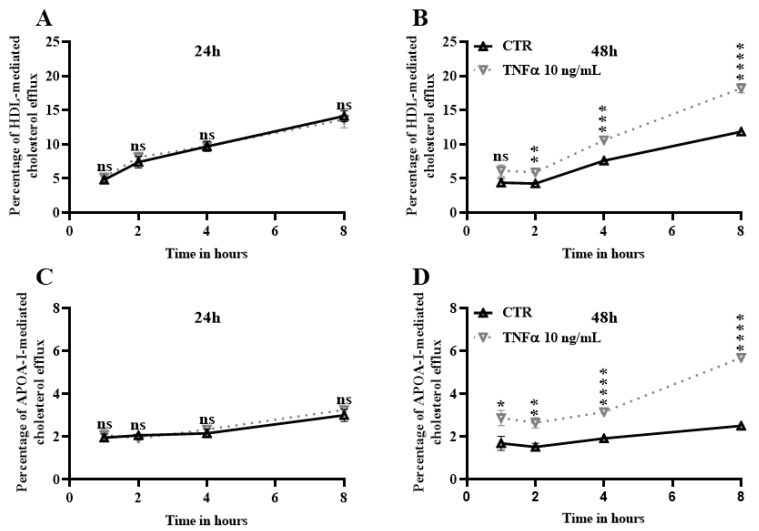
Cholesterol efflux is increased after 48 h of treatment with TNFα. Radiolabeled cholesterol efflux to lipid acceptors was measured after 24 and 48 h of treatment with TNFα (10 ng/mL) during 1, 2, 4, and 8 h. Control condition (CTR) refers to the cells treated only with vehicle as described in the Materials and Methods section. HDL (50 µg/mL)-mediated cholesterol efflux is shown in (**A**,**B**). APOA-I (20 µg/mL)-mediated cholesterol efflux is shown in (**C**,**D**). Each time point represents the mean ± SEM of two independent experiments within three replicates each (N = 2; n = 6). Statistical analysis: Student’s *t*-test, ns: non-significant * *p* < 0.05, ** *p* < 0.01, *** *p* < 0.001, **** *p* < 0.0001.

**Figure 6 ijms-24-05992-f006:**
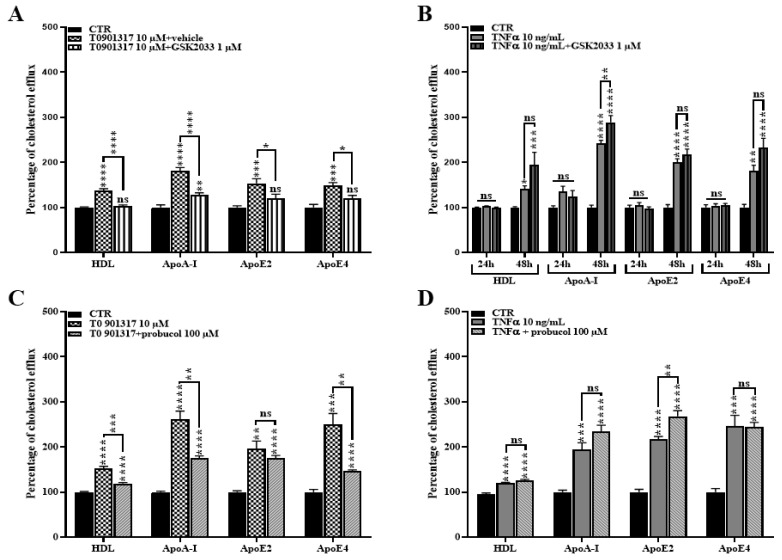
GSK2033 and Probucol partially rescued the T0901317-mediated cholesterol efflux, but not the TNFα-mediated cholesterol efflux. HBP were pre-treated for 30 min with GSK2033 (1 µM) then the *LXR* agonist T0901317 (10 µM) (**A**) was added in the supernatant for 24 h or the TNFα (10 ng/mL) (**B**) for 24 h and 48 h. Cholesterol efflux to lipid acceptors HDL (50 µg/mL), APOA-I (20 µg/mL), APOE2 and E4 (20 µg/mL) was then assessed during 8 h. Each bar represents the mean ± SEM of two independent experiments within four replicates each (N = 2; n = 8). Treated conditions were then compared with the control (CTR) conditions (100%). Statistical analysis: one-way ANOVA followed by Tukey’s multiple comparison test, ns: non-significant, * *p* < 0.05, ** *p* < 0.01, *** *p* < 0.001, **** *p* < 0.0001. Human brain pericytes were treated for 24 h with the *LXR* agonist T0901317 (10 µM) (**C**) and for 48 h with TNFα (10 ng/mL) (**D**). Cholesterol efflux to lipid acceptors HDL (50 µg/mL) and *APOA-I*, *APOE2*, *APOE4* (20 µg/mL) was then assessed during 8 h in the presence or absence of the *ABCA1* inhibitor, Probucol (100 µM). Each bar represents the mean ± SEM of two independent experiments within four replicates each (N = 2; n = 8). Treated conditions were then compared with the control (CTR) condition (fixed at 100%). Statistical analysis: one-way ANOVA followed by Tukey’s multiple comparison (for equal variances) and one-way ANOVA Welsh test followed by Games-Howell’s multiple comparison (for significantly different variances): ns: non-significant, * *p* < 0.05, ** *p* < 0.01, *** *p* < 0.001, **** *p* < 0.0001.

**Figure 7 ijms-24-05992-f007:**
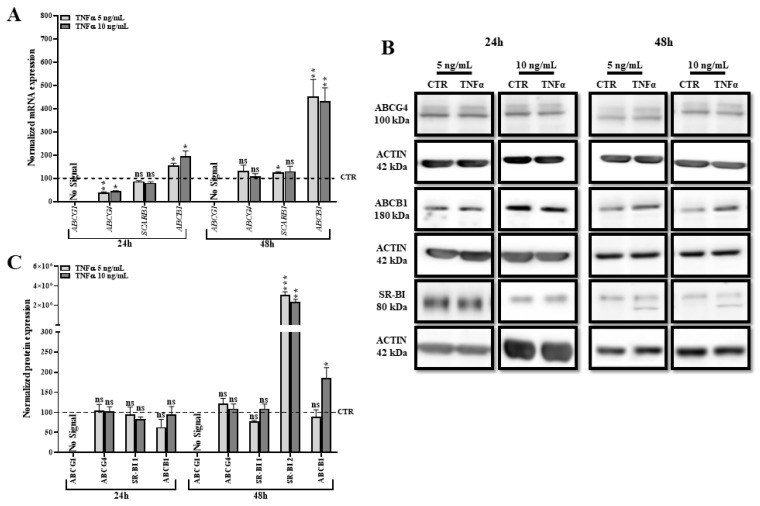
TNFα increases expression of other transporters involved in lipid release. (**A**) Transcriptional expression of *ABCG1*, *ABCG4*, *SCARB1*, and *ABCB1* mRNAs was monitored by RT-qPCR. Each bar represents the level of mRNA normalized to the housekeeping gene *GAPDH*, relative to the control condition (CTR). Data are shown as the mean ± SEM of three experiments from pooled triplicates (N = 3). (**B**) *ABCG4*, *ABCB1* and SR-BI protein levels assessed by Western blot. (**C**) Each bar represents the level of quantified protein signal normalized to ACTIN, relative to the CTR condition. Data are shown as the mean ± SEM of three independent experiments (N = 3). Statistical analysis for (**A**,**C**): Student’s *t*-test, ns: non-significant, * *p* < 0.05, ** *p* < 0.01, *** *p* < 0.001.

**Figure 8 ijms-24-05992-f008:**
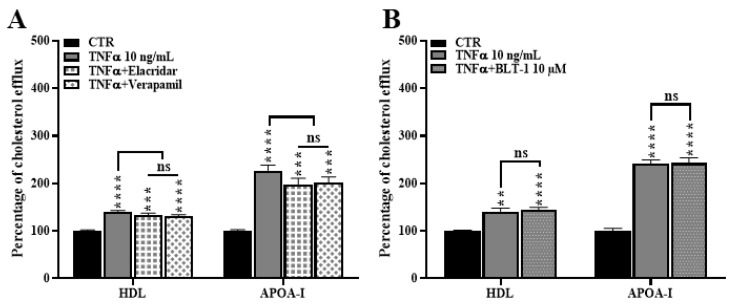
SR-BI and *ABCB1* are not involved in TNFα mediated cholesterol efflux from HBP. After 48 h of incubation with TNFα (10 ng/mL), cholesterol efflux to lipid acceptors HDL (50 µg/mL) and APOA-I (20 µg/mL) was assessed during 8 h in the presence or absence of elacridar (10 µM) or verapamil (50 µM) to inhibit *ABCB1* (**A**) or BLT-1 (10 µM) to inhibit SR-BI (**B**). Each bar represents the mean ± SEM of two independent experiments within four replicates each (N = 2; n = 8). Control condition (CTR) was only treated with vehicle. Statistical analysis: one-way ANOVA followed by Tukey’s multiple comparison (for equal variances) and one-way ANOVA Welsh test followed by Games-Howell’s multiple comparison (for significantly different variances) ns: non-significant, ** *p* < 0.01, *** *p* < 0.001, **** *p* < 0.0001.

**Figure 9 ijms-24-05992-f009:**
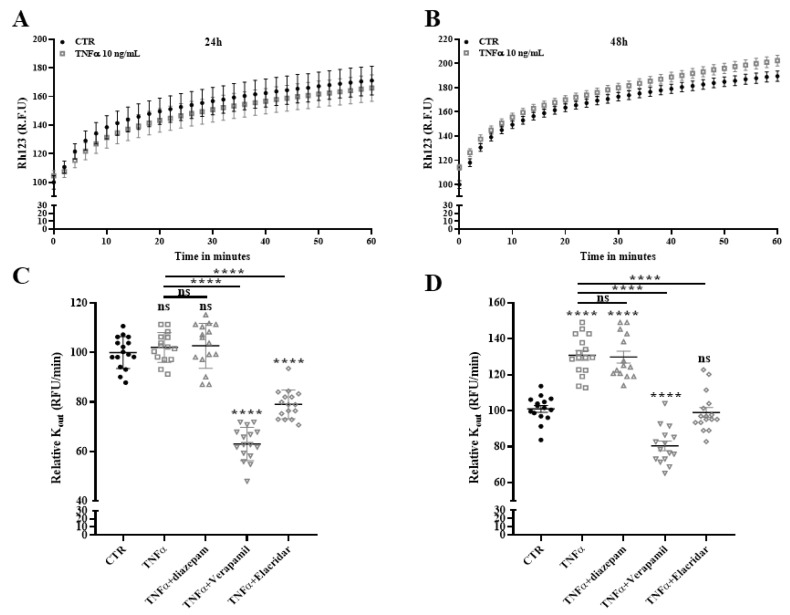
TNFα increases *ABCB1* function in HBP. Rhodamine 123 (Rh 123) is an *ABCB1* substrate and its efflux is monitored in real time during 1 h in TNFα-treated HBP for 24 h (**A**) and 48 h (**B**). Efflux rate (K_out_) of rhodamine 123 was measured in the absence or presence of *ABCB1* inhibitors and calculated as described in the Materials and Methods section (**C**,**D**). K_out_ for control (CTR) condition was fixed at 100% and relative K_out_ of each condition was calculated (CTR 24 h: 42.36 RFU/min ± 0.92; CTR 48 h: 43.95 RFU/min ± 2.22). Each bar represents the mean ± SEM of two independent experiments within eight replicates each (N = 2; n = 16). Statistical analysis: one-way ANOVA followed by Tukey’s multiple comparison: ns: non-significant, **** *p* < 0.0001.

**Figure 10 ijms-24-05992-f010:**
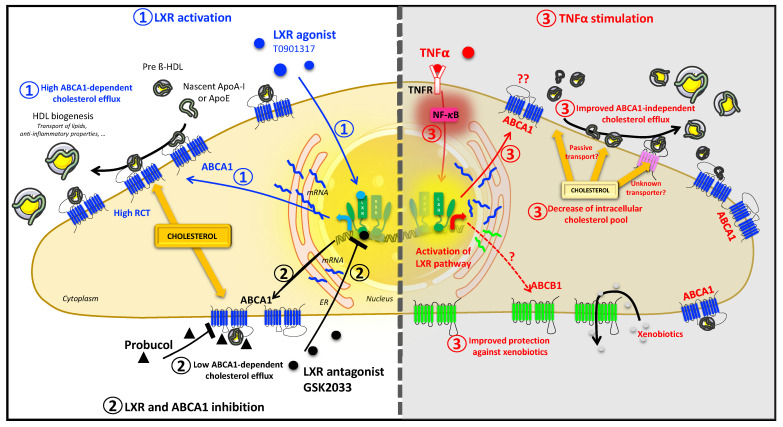
TNFα induces an *LXR*/*ABCA1* independent cholesterol efflux. In HBP, T0901317 (*LXR* agonist) activates the *LXR* pathway which upregulates *ABCA1* gene and protein expression. Consequently, the reverse cholesterol transport (RCT) mediated by *ABCA1* is increased (1). However, *LXR* pathway inhibition via the antagonist GSK2033, suppress *ABCA1* expression and consequently inhibits the T0901317-*ABCA1*-mediated cholesterol efflux (2). The same effect was reported using Probucol, the *ABCA1* inhibitor (2). Under inflammatory stimulation, TNFα activates the NF-κB pathway and the *LXR* pathway (3). As in the T0901317 case, activation of the *LXR* pathway upregulates *ABCA1* and correlates with an increased cholesterol efflux (3). Contrary to the T0901317’s case, neither *LXR* inhibition with GSK2033, nor *ABCA1* inhibition with Probucol, were able to suppress the TNFα-induced cholesterol efflux, suggesting that this efflux is *LXR*/*ABCA1*-independent. The TNFα-induced cholesterol efflux is possibly related to another transporter not identified in our study or to a passive transport, a direct lipid exchange between the plasma membrane and the cell surrounding lipid acceptors. In parallel, TNFα stimulation upregulates *ABCB1* gene and protein expression (3). The activity of this protein is increased, and, consequently, HBP protection against xenobiotics is improved (3). Thus, TNFα induces an *LXR*/*ABCA1*-independent cholesterol efflux and xenobiotics efflux in HBP, shedding light on the possible role of pericytes in diseases progression.

**Table 1 ijms-24-05992-t001:** Primers sequences for qPCR. Primer pairs: F: forward primer and R: reverse primer.

Target	Sequence (F/R)	Accession Number
*ABCA1*	F: CAGTGCTTCCTGATTAGCACAC R: AGGCTAGCGAAGATCTTGGTG	NM_005502.4
*ABCB1*	F: CAGACAGCAGCTGACAGTCCAAGAACAGGACT R: GCCTGGCAGCTGGAAGACAAATACACAAAATT	NM_001348945.2
*ABCG4*	F: GGACATAGAGTTCGTGGAGC R: GGTCTTATAACCCCTTTTGCGCC	NM_001348191.2
*SCARB1*	F: ATCCCCTTCTATCTCTCCGTCT R: GTCGTTGTTGTTGAAGGTGATG	NM_001367988.1
*HMGCR*	F: TGTGTGTGGGACCGTAATGG R: GCTGTCTTCTTGGTGCAAGC	NM_001130996.2
*APOE*	F: GGTCGCTTTTGGGATTACCT R: CCTTCAACTCCTTCATGGTCTC	NM_001302690.2
*LDLR*	F: TTCATGGCTTCATGTACTGGAC R: TTTTCAGTCACCAGCGAGTAGA	NM_000527.5
*LRP1*	F: AATGAGTGTCTCAGCCGCAA R: AACGGTTCCTCGTCAGTCAC	NM_002332.3
*MYLIP*	F: TATGTGACGAGGCCGGACG R: TGATTCCCAGTCGCCTGCAC	NM_013262.4
*NR1H3* *(LXRα)*	F: CAGGGCCATGAATGAGCTGC R: TGTGCTGCAGCCTCTCTACC	NM_005693.4
*NR1H2* *(LXRβ)*	F: TCCTACCACGAGTTCCCTGG R: TGGTTCCTCTTCGGGATCTGG	NM_007121.7
*TNFRSF1A* *(TNFR1)*	F: ACAAGCCACAGAGCCTAGACACTG R: ACGAATTCCTTCCAGCGCAACG	NM_001065.4
*TNFRSF1B* *(TNFR2)*	F: TCTCCAACACGACTTCATCCACGG R: AGACTGCATCCATGCTTGCATTCC	NM_001066.3
*GAPDH*	F: GATGACATCAAGAAGGTGGTGA R: GCTGTTGAAGTCAGAGGAGACC	NM_001357943.2

**Table 2 ijms-24-05992-t002:** Western blot antibody details. Summary of primary antibodies used for Western blots.

Target	kDa	Reference	Supplier	Conditions	Dilution
*ABCA1*	254	ab18180	Abcam	non-heated/reduced	1:1000
*ABCB1 (P-gp)*	180	GTX23364	Genetex	heated/reduced	1:500
*ABCG2 (BCRP)*	72	Ab207732	Abcam	non-heated/reduced	1:1000
*ABCG4*	100	PA5-34855	Invitrogen	heated/reduced	1:1000
*SR-BI*	80	ab52629	Abcam	heated/reduced	1:1000
*HMGCR*	100	Mab90619	Sigma	non-heated/reduced	1:1000
*LRP1*	80	sc57351	Santa Cruz	heated/reduced	1:500
*LDLR*	140	ab52818	Abcam	heated/reduced	1:1000
*APOE*	34	ab1906	Abcam	heated/reduced	1:1000
*COX2*	70	NBD100-689SS	Novus	heated/reduced	1:1000
*NLRP3*	118	ab263899	Abcam	heated/reduced	1:1000
*VCAM1*	90–100	ab134047	Abcam	heated/reduced	1:1000
*IL-6*	22–28	NBD600-1131SS	Novus	heated/reduced	1:1000
*PhosphoNF-κB*	66–75	Mab72261	R&D	heated/reduced	1:1000
*NF-κB*	66–75	ab32536	Abcam	heated/reduced	1:1000
*ACTIN*	42	A5441	Sigma	heated/reduced	1:10,000

## Data Availability

The data that supports the findings of this study are available on request from the corresponding author.

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
