# Peer review of "TNFα Activates the Liver X Receptor Signaling Pathway and Promotes Cholesterol Efflux from Human Brain Pericytes Independently of ABCA1"

_ijms, 2023, doi:10.3390/ijms24065992_

Round 1

Reviewer 1 Report

The work is well presented and the data are clear.

I think it is a good manuscript which must however be improved to provide an adequate conclusion.

Indeed, the conclusion of the abstract is quite disappointing  "In conclusion, our data suggest that inflammation increases HBP protection against xenobiotics and triggers a LXR/ABCA1 independent cholesterol release"

Few additionnal data on the impact of TNF alpha on cholesterol metabolism would permit to give a more precise conclusion and to improve the quality of the manuscript.

My main question is: how can you affirm that this is an independent LXR/ABCA1 cholesterol release; can you summarize in the manuscript, can you present a scheme?

In the manuscript, I think that few additionnal works are required to improve this conclusion and to briefly evaluate complementary hypotheses.

Are there effects of TNFalpha on other patways which would contribute to reduce cholesterol release. Do you have effects of TNFalpha on the intracellular level of cholesterol.

* Can you measure total cholesterol per-cell?

* Using RT-qPCR can you have a look on the expression of MYLIP wich is involved in the control of intracellular cholesterol synthesis, and on the expression of IDOL which regulate LDL-r expression .

Minor remarks:

What is the constitutive level of TNF-receptor, LXR-alpha and beta in human brain pericytes. Can you add these data in supplementary materials.

Reviewer 2 Report

The authors investigate the role of the LXR/ABCA1 axis in human brain pericytes upon stimulation with TNFa. By performing mostly qPCR and Western Blot they show that TNFa increases the cholesterol efflux of these cells, but this process is independent of LXR/ABCA1.

The introduction reads ok. However, I miss the importance of why they study this process in pericytes, I would expect a bigger part on pericytes in the introduction but now they appear a bit out of the blue and in four summarizing sentences. Maybe a lot of the introduction used in the result or discussion section can be described in the introduction instead. Formulating a hypothesis based on the authors earlier work would help the storyline as well.

The presence of LXR/ABCA1 was shown by the last author already in bovine primary pericytes. This time it’s a human pericyte cell line. Although this is an important confirmation, it is not that surprising or novel finding, I would suggest making this confirmational part more central to the storyline.

The experiments seem reasonably sound. At times the figures seem inconsistent in number of repetitions (N=2, N=3, or N=25!) or concentration used (5 or 10 ng/ml TNFa) Also the statistical choices are not fully clear to me, the use of a parametric t-test can be done under a number of assumptions, are these met and how does the data reduction take place?

Overall, I do see the benefit of the paper, but as it stands, I suggest a reappraisal and restructuring of the figures in line of the main message.  Most importantly, I think additional experiments might be necessary to have at least a third independent observation instead of the current two.

Specific general remarks:

-        What is the expression of LXRs (both LXRa and LXRb) in the pericytes?

-        Why keep using both concentrations of TNFa when there is no difference between them?

-        General addition in the method section would be appreciated about growing the pericytes -> they grow in 33C and differentiate in 37C. Also, which passages have been used?

-        It is all in vitro work; a lot of qPCRs, Western blots and cholesterol efllux assays under different conditions. For instance, is ICC possible to investigate the cholesterol content in the pericyte?

Comments per figure:

Figure 1. TNFa promotes expression of inflammatory markers and does not affect human brain pericyte survival

-        Quality of blots is really poor, is this because of the submission system?

-        Cox-2 antibody seems very unspecific.

-        Why is albumin used for normalization IL-6, or is this supposed to be ACTIN as described in the method section? Why not an ELISA for IL-6?

Figure 2. TNFa alters HBP cholesterol metabolism

-        I would include the expression of LXRs (a+b) upon 24 and 48h TNFa treatment.

-        Did you actually do 25 independent experiments, with 96 replicates, where these 96 averaged to one observation for the t-test?

-        Can you report effect size or do you have any other way to interpret the size of the observed effect? From 100% decreased to around 80/90% is minor change.

Figure 4. TNFa increases the expression of ABCA1.

-        Combine with figure 5.

Figure 5. LXR inhibition alleviates TNFa mediated ABCA1 expression.

-        Link LXR-ABCA1, not really surprising.

-        Again, it would be interesting to see what happens with the LXRs under these kinds of stimulations and then in relation to the ABCA1 expression.

Figure 6. TNFa increases cholesterol efflux.

-        What happened to the 5ng/ml TNFa treatment?

-        Maybe highlight the consistency between figures you have.

-        Only N=2, each containing 6 replicates -> 2 independent experiments is not enough, I think. In general, maybe show individual data points so to better understand the significance.

Figure 7. ABCA1 inhibition does not rescue the TNFa induced cholesterol efflux

-        Here ApoA1 higher acceptor than HDL -> different from figure 6.

-        4stars in statistics with just 2 independent experiments sounds strange and a little worrying to me, please explain.

-        Again 5ng/ml is missing, why not use 10ng/ml throughout all figures? Then you can combine them (7&8) and make it more comprehensive.

Figure 8. Probucol rescued the T0 mediated cholesterol efflux but not TNF mediated efflux.

-        Again only 2 independent experiments. To reiterate please clarify wat is tested by the statistics and under which assumptions a parametrical test is chosen and warranted.

-        Combine with figure 7.

Figure 9. TNFa increase expression of other transporters involved in lipid release.

-        5ng/ml TNF is back, why?

Figure 10. SR-BI and P-gp are not involved in TNFa mediated cholesterol efflux in HBP

-        Out of curiosity, did you check what happens to the activity of both receptors when stimulated with the LXR agonist?

Figure 11. TNFa increases p-gp function.

-        Why check activity, seems not connected to the story regarding cholesterol efflux. It’s a finding that is interesting but does not fit.

-        Now back to only 10ng/ml. I see 15 individual datapoints, in 11C and D, what is the experiment, n=1, 15 replicates?

Author Response

In reference to your E-mail of february 21st 2023, please find enclosed our revised manuscript entitled « TNFα activates the Liver X Receptor signaling Pathway and promotes cholesterol efflux from human brain pericytes independently of ABCA1». We made all our possible effort to respect the 14 days of limitation that was given to us to reply, and to address the concerns raised by the 2 reviewers.

We thank the 2 reviewers for their productive criticism and helpful comments. Please be sure that we have made every attempt to address the questions and comments.

As suggested by the reviewers, new experiments have been performed and were added in supplementary data. Corrections have been made. Our responses to their suggested revisions are detailed in our point-by-point reply below. Please find the original remarks by the reviewers written in bold blue letters followed by our reply written in black letters. Modifications in our manuscript appear on red color. These sentences/paragraphs have been copied/pasted in red letters in our responses below to help the reviewers to easily see the modifications.

We are convinced that our manuscript has significantly improved following the reviewer’s suggestions. We hope to have addressed all comments in a satisfactory manner to allow for publication of our revised manuscript in the International Journal of Molecular Sciences.

Reviewer 2

The introduction reads ok. However, I miss the importance of why they study this process in pericytes, I would expect a bigger part on pericytes in the introduction but now they appear a bit out of the blue and in four summarizing sentences. Maybe a lot of the introduction used in the result or discussion section can be described in the introduction instead.

We thank the reviewer for highlighting this point. According to the reviewer comments, we added several information related to the brain pericytes in the introduction of the manuscript. For example, lines 77-78: However, there is a gap in knowledge persisting in the role of brain pericytes in inflam-matory conditions. Lines 79-83: During the last 15 years, advances in high resolution imaging techniques, transcriptomic technics, new cell isolation methods, and cultivation protocols allowed to exponentially increase the studies focusing on brain pericytes [21], thus highlighting new functions for them in CNS homeostasis and BBB functioning.

Formulating a hypothesis based on the authors earlier work would help the storyline as well.

We thank the reviewer for highlighting this suggestion. As suggested, we formulated the hypothesis based on our previous work to improve the storyline.

Lines 91-96: Based on all these observations, and on our previous works obtained in bovine pericytes [26,27], we therefore hypothesized that LXR/ABCA1 axis also controls the cholesterol efflux in human brain pericytes (HBP) and that this process might be affected by inflammation. Using a cell line recapitulating the HBP phenotype [28-30], the objective of our study was to characterize the effects of TNFα signaling on cholesterol transport/metabolism in particular focusing on the LXR/ABCA1 axis.

The presence of LXR/ABCA1 was shown by the last author already in bovine primary pericytes. This time it’s a human pericyte cell line. Although this is an important confirmation, it is not that surprising or novel finding, I would suggest making this confirmational part more central to the storyline.

We thank the reviewer for this comment that will help to promote the work done. We also would like to point out that another important finding of this study is that inflammation triggers the LXR/ABCA1 axis and promotes the cholesterol release independently of ABCA1. As suggested by the reviewer, we highlighted in several part of the manuscript that these data are the first obtained in human brain pericytes demonstrating that LXR/ABCA1 controls the cholesterol efflux when directly activated by agonist but not in inflammatory conditions. Sentences or words in red color were added in introduction, discussion and conclusion.

The experiments seem reasonably sound. At times the figures seem inconsistent in number of repetitions (N=2, N=3, or N=25!) or concentration used (5 or 10 ng/ml TNFa) Also the statistical choices are not fully clear to me, the use of a parametric t-test can be done under a number of assumptions, are these met and how does the data reduction take place?

We thank the reviewer for this comment. As this point is also pointed out by the reviewer for each figure, please find below our response regarding this. We also would like to indicate ti the reviewer that we described our approach in the material and methods section of the manuscript.

Lines 682-697: Results are presented as the mean ± SEM or ± SD as indicated in each figure legends. Normal distribution of values was analyzed using the Shapiro-wilk test. Statistical testing for significance was performed using Student’s t-test or one-way ANOVA test followed by different tests for multiple comparison as indicated in figure legends. All the statistical tests were performed using Prism Software (GraphPad Software Inc., San Diego, CA, USA).

Overall, I do see the benefit of the paper, but as it stands, I suggest a reappraisal and restructuring of the figures in line of the main message.

We thank the reviewer for this comment. According to the reviewer’s comments, we restructured the figures. Figures 5 and 6 were merged as well figures 7 and 8. Legends were merged and the text was updated. New experiments have been performed and a final figure summarizing all the results was provided (Figure 10 of the revised version).

Specific general remarks:

What is the expression of LXRs (both LXRa and LXRb) in the pericytes?

This is an important question that the reviewer pointed out. To answer to this, expression levels of LXRα and LXRβ were analyzed by qRT-PCR in human brain pericytes. Results were added in supplementary data S2. We compared expression of LXRα and LXRβ to GAPDH, and we also included TNFα receptors expression. We observed that LXRα is 3-fold more expressed than LXRβ. The expression of TNFR1 is very high when compared with TNFR2. Figures, legends, table I were updated.

Transcriptional expression of NR1H3 (LXRα), NR1H2 (LXRβ), TNFRSF1A (TNFR1), TNFRSF1B (TNFR2) mRNAs was monitored by RT-qPCR. Each bar represents gene expression relative to the GAPDH (fixed at 100%). Data are shown as the mean ± SEM of three experiments from pooled triplicates (N=3). Statistical analysis: Student’s t-test, **** p <0.0001.

We added Lines 172-175: HBP express both LXR isoforms with LXRα that is expressed almost 3-fold more than LXRβ (Supplementary Material Figure S2). TNFα receptor 1 (TNFRSF1A) is highly expressed whereas expression of TNFα receptor 2 (TNFRSF1B) is very low when compared to GAPDH (Supplementary Material Figure S2).

Why keep using both concentrations of TNFa when there is no difference between them?

We thank the reviewer for this question. In fact, doses used in in vitro studies are always debating because sometimes too low, or sometimes too high. Of course, if the concentration is too low, we can miss the effects. On the contrary, if the concentration is too high, it is possible to create artefact by generating false cellular responses and/or toxicity. In several studies focusing on the effects of TNFα in vitro, in particular in the blood-brain barrier field, the concentrations used vary from 1 ng/mL to 500 ng/mL, but were generally 5 or 10 ng/mL (Da Rocha et al., 2022; Versele et al., 2022). During our study, we selected the doses of 5 ng/mL and 10 ng/mL, expecting to mimic like a dose-response that strengthen the results. This is what we observe for example for ABCA1 because after 24 hours of treatment with 5 ng/mL of TNFα, its expression is not significantly upregulated, but is increased by 2-fold with the treatment of 10 ng/mL (Figure 4C).

For the mechanistic effects (cholesterol quantification and efflux, and pump out assay), after validating that the related targets were not variating differently at 5 ng/mL or 10 ng/mL, we decided to use/show only the 10 ng/mL concentration.

General addition in the method section would be appreciated about growing the pericytes -> they grow in 33C and differentiate in 37C. Also, which passages have been used?

We thank the reviewer for these remarks. In our study, HBP were only cultivated at 37°C. We added these information in the appropriate section. Then, to keep consistency in our study, and to avoid any dedifferenciation issues, we used human brain pericytes at passage 11 for each experiment. For each experiment, a new vial of pericytes was thawed and used, without amplifying or reusing them. All these information was now added in the material and methods section of the manuscript Lines 510-525: “Briefly, these pericytes were transfected and immortalized using retro-virus vectors holding human temperature-sensitive SV40 T antigen (tsA58) and human telomerase (Htert). HBP were then amplified and cultured at 33°C in high glucose (4.5 g/L) Dulbecco’s modified Eagles’ medium (DMEM/HG), supplemented with 10 % non-heat-inactivated Fetal Calf Serum (FCS, Sigma-Aldrich, Saint-Louis, MS, USA), 1 % L-glutamine (Merck chemicals, Darmstadt, Germany) and 1 % penicillin-streptomycin (Sciencell, Carlsbad, CA, USA). Morphology and expression of HBP markers such as desmin, NG2 (neurogli-al2), αSMA (α Smooth Muscle Actin) and PDGF-R (Platelet Derived Growth Factor Recep-tor) were evaluated and HBP were deeply characterized in previous studies.

Before seeding, well plates are coated with collagen I (Corning, NY, USA). Then, HBP were seeded in 6-well plates at a density of 150 000 cells/well, in 12- well plates at a den-sity of 50 000 cells/well, in 24- well plates at a density of 25 000 cells/well or in 96- well plates at 5 000 cells/well for pump out assays and 7 500 cells/well for LXR cignal reporter assay. Only brain pericytes at passage 11 were used in this study. Cells were cultured at 37°C in DMEM/HG supplemented with 1 % L-glutamine (Merck KGaA, Darmstadt, Ger-many) and 1% penicillin–streptomycin (Sciencell) during 48 h to reach 80 to 90 % of con-fluence.”

It is all in vitro work; a lot of qPCRs, Western blots and cholesterol efflux assays under different conditions. For instance, is ICC possible to investigate the cholesterol content in the pericyte?

We thank the reviewer for this suggestion and we added this limitation of our study in the last part of the discussion section of our manuscript.

Lines 497 to 499: “Importantly, we generated these data in HBP cultured in vitro. Further studies with human samples or tissues are needed to confirm these effects of the inflammation at the human BBB and CNS.”

The experiments seem reasonably sound. At times the figures seem inconsistent in number of repetitions (N=2, N=3, or N=25!) or concentration used (5 or 10 ng/ml TNFa) Also the statistical choices are not fully clear to me, the use of a parametric t-test can be done under a number of assumptions, are these met and how does the data reduction take place?

We apologize that the reviewer did not understand the inconsistency in number of repetition and the statistical choices. Please find below our explanations that, we hope, will clarify everything.

First, we worked with different formats for the different kind of experiments that we performed. As we indicated in the material and methods, human brain pericytes were seeded in 6-well plates at a density of 150 000 cells/well (for Western blots), in 12- well plates at a density of 50 000 cells/well (for RT-qPCR and cholesterol time curve efflux experiments), in 24- well plates at a density of 25 000 cells/well (for cholesterol efflux experiments, Resazurin tests) or in 96- well plates at 5 000 cells/well for pump out assays and 7 500 cells/well for LXR cignal reporter assay. These different formats make possible to perform each condition in 3, 4 or 8 different wells, thus resulting, for example, in N=2, but n=8 (eight different treated and analyzed wells in 2 independent experiments). We therefore analyzed these 8 wells, and made a mean with these 8 values. Then, these 8 values were used for statistical analysis. As mentioned in our material and methods section (4.11), we performed normality test using the Shapiro-Wilk test. As also mentioned in the figures legends, we used an ANOVA test, followed by Tukey’s multiple comparison (for equal variances) and one-way ANOVA Welsh test followed by Games-Howell’s multiple comparison (for significantly different variances).

Comments per figure:

Figure 1. TNFa promotes expression of inflammatory markers and does not affect human brain pericyte survival

-        Quality of blots is really poor, is this because of the submission system?

We apologize for this issue meet by the reviewer. We provided pictures at 400 dpi following the guidelines of the journal. Indeed, this issue probably raised by the submission system. For sure, if better picture quality is requested by the journal, we will make our best to provide highest quality.

Cox-2 antibody seems very unspecific.

We totally agree with the referee. When we tested several concentrations of TNFα, we observed that only this band, localized at almost 70kDa, is dose dependently increased. Other bands remain unchanged demonstrating that these signals are not specific. The objective to investigate COX-2 in our study was to prove that our TNFα treatments trigger an inflammatory response. To support this, we checked other different targets: VCAM-1, NLRP3 and IL-6. All these inflammatory markers, including the 70kDa band corresponding to COX-2, increase thus confirming that TNFα promotes inflammatory responses in human brain pericytes. So despite the poor specificity of the used antibody we are convinced that this is informative for our study.

Why is albumin used for normalization IL-6, or is this supposed to be ACTIN as described in the method section? Why not an ELISA for IL-6 ?

We thank the reviewer for this question. We gently remind the reviewer that IL-6 and APOE are two secreted proteins. As mentioned in the figure’s 1 legend, IL-6 is detected in the supernatant after protein precipitation and not from cell lysates. Therefore, it is not possible to normalize IL-6 using ACTIN which is an intracellular protein. Also, as mentioned in our materials and methods, TNFα treatments are made in media containing 0.1% of BSA (Bovine Serum Albumin), so when we precipitate IL-6 from the supernatant we use the albumin as a loading control for normalization as we already did in our previous study (Lamartinière et al., 2018). To avoid any misunderstanding, we will add more details in the material and methods section as follow :

“4.6 APOE and IL-6 determination in cell culture supernatant.

Supernatants were collected after each TNFα treatment and stored at -20°C. To precipitate secreted proteins, 4 volumes of pure acetone (Sigma-Aldrich) were added to one volume of supernatant and incubated at -20°C overnight. After a centrifugation step at 10 000 g for 10 minutes, supernatant was discarded and remaining acetone was air-dried. The pellet was resuspended in RIPA lysis buffer supplemented with anti-proteases and anti-phosphatases. Bradford assay was performed for protein quantification and 50 µg of proteins were used for western blot analysis. Then, immunoblots were performed as described above. Secreted targets were normalized to BSA.”

Figure 2. TNFa alters HBP cholesterol metabolism

-        I would include the expression of LXRs (a+b) upon 24 and 48h TNFa treatment.

We agree with the reviewer that this information might be useful for our study. Because we only had 14 days to reply to all the reviewers’ comments, it was not possible to purchase the antibodies and to perform/optimize the Western blots experiments. However, we performed RT-qPCR because we had previously designed the primers. We therefore performed RT-qPCR and added the results in supplementary figure S2B.

Transcriptional expression of LXRα and LXRβ mRNAs was monitored by RT-qPCR. Each bar represents the level of mRNA normalized to the housekeeping gene GAPDH, relative to the control condition (CTR). Data are shown as the mean ± SEM of three experiments from pooled triplicates (N=3). Statistical analysis: Student’s t-test, ns: non-significant, *p < 0.05.

As shown by the results above, a slight but significant upregulation of LXRβ expression was observed after 48 hours of treatment with 10 ng/mL of TNFα. No changes in LXRα expression have been observed. This data was added in the supplementary figure S2B and described/discussed in the manuscript (Lines 179-181 and lines 411-412).

-        Did you actually do 25 independent experiments, with 96 replicates, where these 96 averaged to one observation for the t-test?

For each cholesterol efflux experiment, we also quantified the intracellular cholesterol level. These experiments are shown in the new figures 5, 6, and 8. In total, we therefore performed 25 independent experiments detailed below:

Fig 5: Cholesterol efflux kinetics done with HDL (N=2; n=6) and APOA-I (N=2; n=6). Quantification of intracellular cholesterol in total: N=4; n=12.

Fig 6A/B: Cholesterol efflux done with HDL (N=2; n=8), APOA-I (N=2; n=8), APOE2 (N=2;n=8), APOE4 (N=2;n=8): Quantification of intracellular cholesterol in total: N=8;n=32.

Fig 6C/D: Cholesterol efflux done with HDL (N=2; n=8), APOA-I (N=2; n=8), APOE2 (N=2; n=8), APOE4 (N=2; n=8): Quantification of intracellular cholesterol in total: N=8; n=32

Fig 8: Cholesterol efflux done with HDL (N=2;n=8), APOA-I(N=2;n=8): Quantification of intracellular cholesterol in total: N=4;n=16

The Last N=1; n=4 is from a kinetic for cholesterol efflux we did with APOE2 and was not shown in the manuscript. Please find these data below:

Not shown in our study : Figure x Cholesterol efflux in presence of APOE2 (20 µg/mL) is increased after 48 hours of treatment with TNFα. Radiolabeled cholesterol efflux to APOE2 was measured after 24 and 48 hours of treatment with TNFα (10 ng/mL) during 1, 2, 4, and 8 hours. Control condition (CTR) is the cells treated only with vehicle as described in material and methods section. Each time point represents the mean ± SD of one experiment within 4 replicates each (N=1; n=4).

Therefore, we calculated in total the intracellular cholesterol level in CTR vs TNFα treated HBP in 25 experiments with a total of 96 replicates. The statistical test was applied to a total of 96 values and performed as described in the legends figures.

Can you report effect size or do you have any other way to interpret the size of the observed effect? From 100% decreased to around 80/90% is minor change.

Cholesterol metabolism is very important for cell physiology and survival. Low variation in cholesterol metabolism can have huge effects on cell physiology. In studies focusing on the cholesterol metabolism, decreases to 80/90% of intracellular content or increases to 110/120% are observed. For example, depleting endothelial cells with ABCA7 expression increases the cell cholesterol content by 26 ± 9 % (Lamartinière et al., 2018). When cultured with lipid depleted medium, endothelial cells show a decrease of 10 % in cholesterol content when compared with cells cultured in fetal calf serum (Fowler et al., 2022).

In this present study, we report a decrease of 20% in the intracellular cholesterol after TNF-α treatment. According to the statistical test, it is significant with ** p < 0.01.

This result was supported in our study by an increase in the cholesterol efflux (Figure 5). As we mentioned in the discussion, this efflux could be related to a change in the plasma membrane composition. Thus, further investigations are needed to understand which cholesterol pool among the three mentioned pools is affected by TNFα treatment.

Figure 4. TNFa increases the expression of ABCA1.

-        Combine with figure 5.

We thank the reviewer for this comment. As suggested by the reviewer, figures 4 and 5 were combined. All figures numbers and order have been updated.

Figure 5. LXR inhibition alleviates TNFa mediated ABCA1 expression.

Link LXR-ABCA1, not really surprising.

We agree with the reviewer that the link between LXR and ABCA1 was already known, and that function of ABCA1 in cholesterol efflux is very well established. However, this LXR/ABCA1 link has never been previously established in human brain pericytes, and under TNFα stimulation.

Our results clearly state for the first time that TNFα increases ABCA1 expression via the LXR signaling pathway. In addition, by adding the positive control T0901317, we also clearly show that our inhibitors (GSK2033 and Probucol) work and that our experimental set-up is adapted.

Again, it would be interesting to see what happens with the LXRs under these kinds of stimulations and then in relation to the ABCA1 expression.

We thank the reviewer for this comment. ABCA1 is the main target gene of the LXR signaling pathway. GSK2033 is an LXR inhibitor and T0901317 is a LXR agonist. Therefore, ABCA1 expression is closely linked to LXR levels and activation/inhibition. We therefore would like to only focus on ABCA1 expression rather to LXR levels that in all cases, will regulate ABCA1 and will result in up- or down-regulation of ABCA1 expression. The aim of our study is to link LXR activation/inhibition to the level of ABCA1, and then to the cholesterol efflux.

Figure 6. TNFa increases cholesterol efflux.

What happened to the 5ng/ml TNFa treatment?

We thank the reviewer for this comment. Our previous results showed that 10 ng/mL has most important effect on ABCA1 expression (Figure 4). Because our objective is to better understand the links between ABCA1 and cholesterol efflux, we then only focused on 10 ng/mL for the cholesterol efflux experiments. These experiments generate a lot of radiolabeled waste products (and we have some annual limitations) and need some special authorizations to be performed in our academic laboratory.

Only N=2, each containing 6 replicates -> 2 independent experiments is not enough, I think. In general, maybe show individual data points so to better understand the significance.

We thank the reviewer for this comment. As explained above, we used the 6 replicates values to calculate the mean and then to make our statistical analysis. All these information are indicated in the material and section as well in the figures legends.

Figure 7. ABCA1 inhibition does not rescue the TNFa induced cholesterol efflux

Here ApoA1 higher acceptor than HDL -> different from figure 6.

We thank the reviewer for this comment. The figures 5 and 6 (new numbering) show the same trend, HDL is a better cholesterol acceptor than APOA-I in basal condition, as previously observed in murine, porcine or bovine cells (Lamartinière et al., 2018; Saint-Pol et al., 2012 and 2013; Panzenboeck et al., 2002, etc) but also after TNFα treatment. For example, in figure 5, after 8 hours of cholesterol release without TNFα, the efflux to APOA-I and to HDL is 2% versus 11%, respectively. In presence of TNFα, the cholesterol efflux increase at 5% for APOA-I and 17% for HDL. HDL remain a better cholesterol acceptor than APOA-I in inflammatory conditions. However, we presented the Figure 6 in relative cholesterol efflux and fixed the control condition (without TNFα treatment) at 100 %. Therefore, cholesterol efflux is increased by 2.5-fold to APOA-I (2% to 5%) versus 1.5-fold to HDL (10% to 17%) giving the feeling that cholesterol efflux is more prone to APOA-I than HDL. But cholesterol release remains highest to HDL (17%) than to APOA-I (5%). Hope that these explanations convinced the reviewer that figures 5 and 6 show the same trend, HDLs are better cholesterol acceptors than APOA-I.

4 stars in statistics with just 2 independent experiments sounds strange and a little worrying to me, please explain.

We thank the reviewer for this comment. As mentioned in the figures legends, our radioactive efflux experiments were done 2 independent times with 4 biological replicates each. So in total we got 8 values for both experiments. These 8 values were used for statistical analysis. As also mentioned in the figures legends, we used a ANOVA test, followed by Tukey’s multiple comparison (for equal variances) and one-way ANOVA Welsh test followed by Games-Howell’s multiple comparison (for significantly different variances). And our results were significant with **** for a p value < 0.0001.

Again 5ng/ml is missing, why not use 10ng/ml throughout all figures? Then you can combine them (7&8) and make it more comprehensive.

We thank the reviewer for this suggestion. As suggested by the reviewer, figures 7 and 8 were combined. Text and legends were changed accordingly. The absence of 5 ng/mL has already been addressed previously.

Figure 8. Probucol rescued the T0 mediated cholesterol efflux but not TNF mediated efflux.

Again only 2 independent experiments. To reiterate please clarify wat is tested by the statistics and under which assumptions a parametrical test is chosen and warranted.

Combine with figure 7.

We thank the reviewer for this suggestion. As suggested by the reviewer, figures 7 and 8 were combined. Text and legends were changed accordingly. As mentioned in the 4.11 section for statistical analysis, we tested the normal distribution of our values with the Shapiro-Wilk test. Then we applied Student’s t-test to compare CTR vs TNFα treated HBP. When we used inhibitors, we applied the One-way ANOVA test followed by multiple comparison tests, to compare CTR, TNFα and TNFα+inhibitor conditions.

Figure 9. TNFa increase expression of other transporters involved in lipid release.

-        5ng/ml TNF is back, why?

As mentioned previously, we checked our targets expression for both concentrations when we do not observe any difference, we use only one concentration (10 ng/mL) for the mechanistic effect and to avoid to generate too much radiolabeled waste products (we have annual limitations).

Figure 10. SR-BI and P-gp are not involved in TNFa mediated cholesterol efflux in HBP

- Out of curiosity, did you check what happens to the activity of both receptors when stimulated with the LXR agonist?

We thank the reviewer for this suggestion. Indeed, we did not check SR-BI and P-gp functions after TO901317 treatment because we focused on effects of inflammation in LXR/ABCA1 axis. However, we started a new study in which we will investigate the effect of other LXR agonists (named oxysterols such as 24S-hydroxycholesterol or 27-hydroxycholesterol) on P-gp expression/function and we planned to add a new condition (TO901317) following the reviewer’s suggestion. Therefore, these results will be included in our next study.

Figure 11. TNFa increases p-gp function.

- Why check activity, seems not connected to the story regarding cholesterol efflux. It’s a finding that is interesting but does not fit.

We thank the reviewer for this important point. We observed that TNFα increases P-gp expression (New Figure 7) but P-gp inhibition does not decrease cholesterol efflux (New Figure 8). So, P-gp is not involved in this cholesterol efflux. However, we performed the experiments of the new Figure 9 for two reasons:

- First, we had no evidence that P-gp inhibition by Elacridar or verapamil correctly worked in figure 8. The results of the figure 9 definitively confirm that P-gp activity is correctly inhibited by these molecules.

- Secondly, we wanted to know if P-gp activity would be increased because its expression is increased. And this is what we confirm with the results of the new figure 9.

Therefore, despite the fact that we understand the point of view of the reviewer, we would prefer to keep the figure 11 (now named Figure 9) in the manuscript. To make this finding more suitable for the reviewer, we added these following sentences in the manuscript:

Lines 359-362 : These results clearly demonstrate that Elacridar and Verapamil also used in Figure 8, efficiently inhibit P-gp functions, that strengthens our previous conclusion that P-gp is not involved in cholesterol efflux after TNFα stimulation

- Now back to only 10ng/ml. I see 15 individual datapoints, in 11C and D, what is the experiment, n=1, 15 replicates?

We thank the reviewer for this question. Why 5 ng/mL is absent was explained previously. For the values number, we forgot to add in the figure legends that this is 2 independent experiments, performed and analyzed in 8 independent wells each. We sincerely apologize for this. Therefore, this is N=2, n=16 that we should add in the legend of the figure 11. This information has been added in the legend of the figure 11 (Now figure 9).

“Each bar represents the mean ± SEM of two independent experiments within 8 replicates each (N=2; n=16).”

Round 2
